# Eagle and Finch: RWKV with Matrix-Valued States and Dynamic Recurrence

**Bo Peng** *
EleutherAI,
RWKV Project (Linux
Foundation AI & Data)
pengbo863@gmail.com

**Daniel Goldstein** *
EleutherAI,
Recursal AI
dan@recursal.ai

**Quentin Anthony** *
EleutherAI,
Ohio State University,
Zyphra
quentin@eleuther.ai

**Alon Albalak**
EleutherAI,
UC Santa Barbara,
SynthLabs

**Eric Alcaide**
EleutherAI,
Charm Therapeutics,
Dalle Molle Institute
for AI Research

**Stella Biderman**
EleutherAI

**Eugene Cheah**
EleutherAI,
Recursal AI,
RWKV Project (Linux
Foundation AI & Data)

**Teddy Ferdinan**
Wroclaw Tech

**Haowen Hou**
Guangdong
Laboratory of
AI and Digital
Economy (SZ)

**Kranthi Kiran GV**
EleutherAI,
New York University

**Satyapriya Krishna**
Harvard University

**Ronald McClelland Jr.**
EleutherAI,
Ronsor Labs

**Niklas Muennighoff**
Contextual AI

**Fares Obeid**
EleutherAI,
Imperial College

**Atsushi Saito**
EleutherAI,
Nextremer Co. Ltd.

**Guangyu Song**
EleutherAI

**Haoqin Tu**
UCAS †,
UC Santa Cruz

**Ruichong Zhang**
Tsinghua University

**Bingchen Zhao**
University of Edinburgh

**Qihang Zhao**
LuxiTech Co. Ltd.

**Jian Zhu**
University of British Columbia

**Rui-Jie Zhu**
University of California, Santa Cruz

## Abstract

We present Eagle (RWKV-5) and Finch (RWKV-6), sequence models improving upon the RWKV (RWKV-4) (Peng et al., 2023) architecture. Our architectural design advancements include multi-headed matrix-valued states and a dynamic recurrence mechanism that improve expressivity while maintaining the inference efficiency characteristics of RNNs. We introduce a new multilingual corpus with 1.12 trillion tokens and a fast tokenizer based on greedy matching for enhanced multilinguality. We trained four Eagle models, ranging from 0.46 to 7.5 billion parameters, and two Finch models with 1.6 and 3.1 billion parameters and find that they achieve competitive performance across a wide variety of benchmarks. We release all our models on HuggingFace under the Apache 2.0 license.

---

*Equal first authorship. Others listed alphabetically.
†University of Chinese Academy of Sciences

# 1 Introduction

Advancements in Large Language Models (LLMs) have significantly impacted Natural Language Processing (NLP) tasks. The field has traditionally been dominated by the transformer architecture (Vaswani et al., 2023). However, the expressive attention mechanism of transformers leads them to suffer from quadratic time complexity with respect to input sequence length. Various methods have been proposed to achieve sub-quadratic time complexity without significantly changing the core attention mechanism, typically relying on some form of sparsity techniques (Child et al., 2019a; Beltagy et al., 2020; Zaheer et al., 2020).

Recent works have achieved sub-quadratic time complexity without significantly sacrificing performance by introducing new mechanisms to replace attention at the core of the Transformer architecture. These models include gated recurrences (Fu et al., 2023; Gu & Dao, 2023; Gu et al., 2021; Sun et al., 2023; Katsch, 2023; Qin et al., 2023; Smith et al., 2023a), gated convolutions (Poli et al., 2023; Peng et al., 2023), and data-dependent linear attention (Yang et al., 2023; Katharopoulos et al., 2020b), sparse attentions (Tay et al., 2020; Child et al., 2019b; Zaheer et al., 2020; Qiu et al., 2019) and their combinations (De et al., 2024; Qin et al., 2024; 2022b). We build off RWKV-v4 introduced in (Peng et al., 2023), which provides efficient inference and training along with a parallelizable implementation compared to competing architectures as shown in Table 1.

| Architecture | Inference | | Parallel | Training | |
|---|---|---|---|---|---|
| | Time | Memory | | Time | Memory |
| LSTM/LMU | $O(1)$ | $O(1)$ | ✗ | $O(N)$ | $O(N)$ |
| Transformer | $O(N)$ | $O(N)^a$ | ✓ | $O(N^2)$ | $O(N)^b$ |
| Linear Transformer | $O(1)$ | $O(1)$ | ✓ | $O(N)$ | $O(N)$ |
| H3/S4 | $O(1)$ | $O(1)$ | ✓ | $O(N \log N)$ | $O(N)$ |
| Hyena | $O(N)$ | $O(N)$ | ✓ | $O(N \log N)$ | $O(N)$ |
| RWKV/Mamba/RetNet | $O(1)$ | $O(1)$ | ✓ | $O(N)$ | $O(N)$ |

Table 1: Comparative analysis of RWKV-4/5/6 and other LLM architectures regarding time and memory complexity for both inference per token and training per sequence, and training parallelizability across the sequence dimension. The context/sequence length is denoted by $N$. $^aO(1)$ without KV cache $^b$With Flash Attention

In this paper, we introduce two new architectures: **Eagle** (RWKV-5) and **Finch** (RWKV-6). First, Eagle improves upon the architecture and learned decay schedule from RWKV-4 (Peng et al., 2023) through the use of expressive multi-headed matrix-valued states (as opposed to vector-valued states), a reformulated receptance, and an additional gating mechanism. Finch further improves the expressivity and flexibility of the architecture by introducing new data-dependent functions for both the time-mixing and token-shift modules, consisting of parameterized linear interpolations. Additionally, Finch proposes a novel use of the Low Rank Adaptation (Hu et al., 2022) function to allow for trainable weight matrices to efficiently augment the learned data decay vectors in a context-dependent manner. Finally, we introduce a new tokenizer, the RWKV World Tokenizer, and a new dataset, RWKV World v2 (1.12 trillion tokens), manually designed to improve performance on multilingual and code data.

Through extensive experimentation, we show that the Eagle and Finch models perform competitively, or improve upon existing models under a wide variety of sequence modeling domains and tasks. Specifically, we evaluate our trained models on commonly used English-only and multilingual text benchmarks, associative recall, music modeling, and vision-language benchmarks. Our experiments demonstrate that the advancements in Eagle and Finch provide significant progress towards developing more efficient AI models.

In summary, our main contributions are:

- The Eagle (RWKV-5) and Finch (RWKV-6) RWKV architectures, which significantly improve over RWKV-4 on benchmarks for LLMs.

- The RWKV World Tokenizer which contains underrepresented languages' vocabulary and which performs fast tokenization with Trie-based greedy matching.
- The RWKV World v2 public dataset, comprised of 1.12 trillion tokens of publicly available multilingual data.
- Public release of four pre-trained Eagle models, scaling from 0.46 to 7.5 billion parameters, and two Finch models, with 1.6 and 3.1 billion parameters. Demonstrating that these novel architectures are competitive to transformers when trained using enough FLOPs to make meaningful scaling conclusions.
- A completely open training pipeline to enable interpretability and reproducibility of alternative-architecture LLMs (See Table 2).

| Model | Context Length | Training Tokens | Open Weights | Open Code | | Open Dataset |
| --- | --- | --- | --- | --- | --- | --- |
| | | | | Inference | Training | |
| GPT-4 | 128k[a] | Undisclosed | ○ | ○ | ○ | ○ |
| LLaMA2 7B | 4k | $2.0 \times 10^{12}$ | ◐ | ● | ○ | ○ |
| Mistral 7B v0.1 | 32k[b] | Undisclosed | ● | ● | ○ | ○ |
| Gemma 7B | 8k | $6.0 \times 10^{12}$ | ◐ | ● | ● | ○ |
| StableLM 7B v2 | 4k | $1.1 \times 10^{12}$ | ● | ● | ● | ● |
| Pythia 6.9B | 2k | $3.3 \times 10^{11}$ | ● | ● | ● | ● |
| Eagle 7B | Indefinite[c] | $1.1 \times 10^{12}$ | ● | ● | ● | ● |

Table 2: Comparison of the openness and accessibility of public foundational LLMs with 7B+ parameters regarding model weights, official inference/training code, and dataset. Widely available but not under an open source license is indicated by ◐.

[a]OpenAI's gpt-4-0125-preview model [b]With sliding window attention [c]Pretrained with context length 4096, but no fundamental context length limitation or relationship to speed, see G.2 for extrapolation details

## 2 Background

Eagle and Finch are RNNs based on a multi-headed hybridization of the RWKV-4 architecture and linear attention. We discuss related work and the evolution of these two architectures below, with a more detailed review given in Appendix C.

Recurrent Neural Networks (RNNs) are well suited to provide inexpensive inference on sequence modelling tasks, typically operating in $O(1)$ time complexity per step with respect to sequence length. They model sequences with time dependencies by generating a hidden state $h_t$ at each time step, which is fed back in at the next time step as a secondary input. Classic RNNs (e.g. LSTM (Hochreiter & Schmidhuber, 1997) and GRU (Cho et al., 2014)) became widely used for sequence modelling, but are difficult to parallelize across the time dimension for training.

The Transformer architecture has enjoyed remarkable success in generative modelling, particularly language modelling (Vaswani et al., 2023; Radford et al., 2018), providing SOTA performance across many tasks. However, the use of multi-headed dot-product self-attention (MHA) leads to quadratic time complexity with respect to sequence length. The deficiencies of classic RNNs and Transformers led to many attempts to develop architectures incorporating the best features of both in a single model, namely $O(1)$ time per token and fast highly parallelizable training.

Linear Attention (Schmidhuber, 1992; Katharopoulos et al., 2020a) replaces the numerator of MHA's $\text{softmax}(QK^T)V$ with $\phi(Q)\phi(K)^{\mathrm{T}}V$, allowing a reordering of operations via associativity to $\phi(Q)(\phi(K)^{\mathrm{T}}V)$, where $\phi$ represents a non-negative feature-map function. It can be computed as an RNN in $O(1)$ time per step by adding $\phi(K_i^T)V_i$ to a recurrent state at each time step $i$, or trained in parallel much like MHA. This accomplishes the main goals outlined above, but naive linear attention suffers from significantly reduced performance compared to MHA-based transformers.

A modified form of linear attention, the Attention Free Transformer (AFT) (Zhai et al., 2021), paved the way for the RWKV architecture, by using a number of attention heads equal to the size of the feature dimension and incorporating a set of learned pairwise positional biases, denoted as $w$.

$$\text{AFTAttn}_t = \sigma_q(q_t) \odot \frac{\sum_{i=1}^{t} \exp(k_i + w_{i,t}) \odot v_i}{\sum_{i=1}^{t} \exp(k_i + w_{i,t})} \tag{1}$$

RWKV-4 reformulates the AFT equation by replacing the pair-wise positional biases with a channel-wise vector of additive weight decay rates $w$. It also adds a bonus term $u$ to offset the weight of the current input specially.

$$wkv_t = \frac{\sum_{i=1}^{t-1} \exp(-(t-1-i)w + k_i) \odot v_i + \exp(u + k_t) \odot v_t}{\sum_{i=1}^{t-1} \exp(-(t-1-i)w + k_i) + \exp(u + k_t)}. \tag{2}$$

RWKV-4 also adds token-shift and gating to both attention and feed-forward sub-blocks of transformer, and small embedding initialization and normalization to quickly arrive at well-distributed token embeddings. Combining all of these architectural changes led RWKV-4 to become the first RNN to rival the performance of Transformers, while maintaining fast parallelizable training and $O(1)$ time complexity per token.

There has been a recent revival of RNNs in NLP research (Tiezzi et al., 2024). HGRN(Qin et al., 2023) is a recent time-parallelizable data-dependent RNN that employs input and forget gates. TransNormer(Qin et al., 2022a) applies RMSNorm to linear attention to bound its output. Other new time-parallelizable data-dependent RNNs have also been invented concurrently with our work including GLA (Yang et al., 2023) and Griffin (De et al., 2024).

State Space Models (SSMs) employ a hidden state of basis function weights to model an approximation of the input function (Gu et al., 2020), updating that hidden state via a differential equation. Earlier SSMs (Gu et al., 2022) were historically computed using long convolutions in $O(N \log N)$ time per sequence, but could also be formulated as a recurrent network. Recently, it has been shown that SSMs can be parallelized across the time dimension via techniques including associative scan (Smith et al., 2023b). A new class of SSMs has also emerged concurrently with our work (Katsch, 2023; Gu & Dao, 2023) that feature data-dependent $A$ and $B$ terms, which function similarly to the data-dependent dynamic recurrence used in Finch.

## 3   Eagle/Finch Architecture

We refine the RWKV architecture in two steps, and observe significant modeling improvements with each. Compared to the baseline RWKV-4, Eagle adds matrix-valued attention states, LayerNorm over the attention heads, SiLU attention gating, and improved initialization. It also removes the Sigmoid activation of receptance. Finch further applies data-dependence to the decay schedule and token-shift.

The core architecture remains similar to that of RWKV-4, consisting of a series of stacked residual blocks shaped like a traditional Transformer. Following notation from (Tolstikhin et al., 2021), each block contains one Pre-LayerNorm Time-Mixing sub-layer followed by one Pre-LayerNorm Channel-Mixing sub-layer, as depicted in Figure 1, left. These correspond to the traditional Attention and Feed Forward Network sub-layers of the Transformer. See Appendix B for more details on our training implementation and the differences from RWKV-4, and Appendix L for speed and memory benchmarks.

## 4   Method

In this section, we use $D$ to denote the model dimension, and unless explicitly stated, all vectors appearing in this section are dimension $D/h$, where $h$ denotes the number of heads,

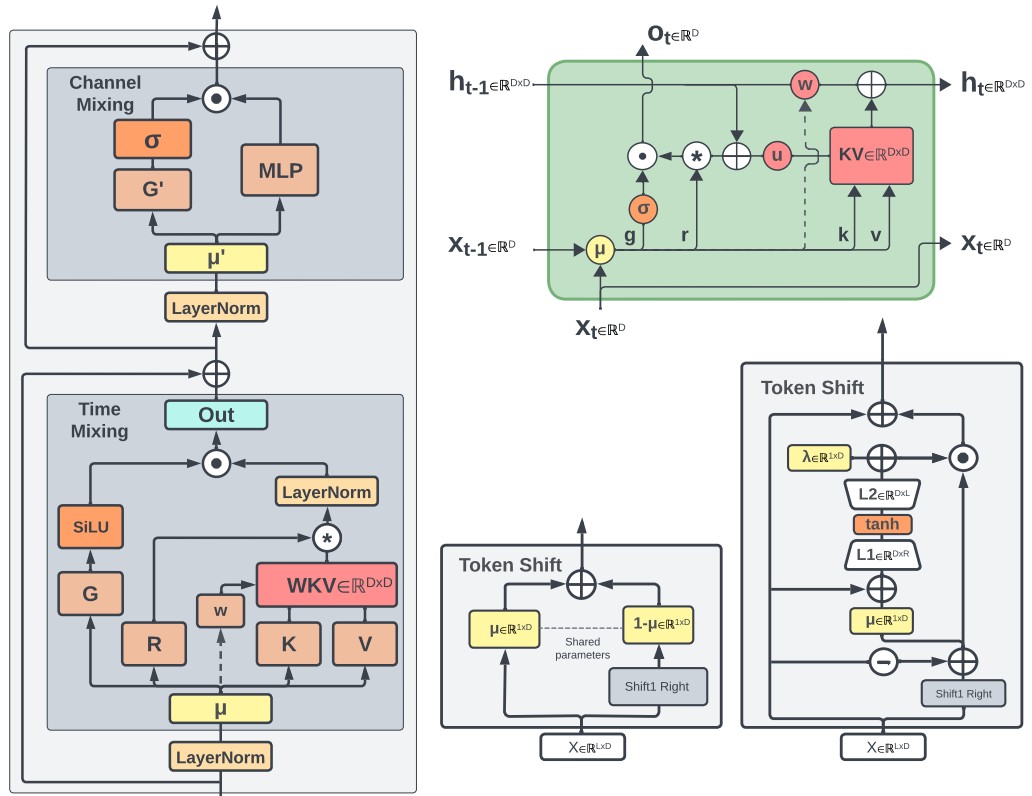

Figure 1: RWKV architecture overview. **Left:** time-mixing and channel-mixing blocks; **top-right:** RWKV time-mixing block as RNN cell; **center-bottom:** token-shift module in FeedForward module and Eagle time-mixing; **bottom-right:** token-shift module in Finch time-mixing. All shape annotations assume a single head for simplicity. Dashed arrows (left, top-right) indicate a connection in Finch, but not in Eagle.

belonging to $\mathbb{R}^{(D/h)}$. For simplicity we show calculations per-head, eliding the head index. We use the convention that all vectors are row vectors unless explicitly transposed, so all matrices operate on the right side. We use the square subscript to denote a variable.

## 4.1 Eagle

**Eagle Token Shift** We adopt the Token Shift technique from the previous RWKV, similar to a 1D causal convolution of size = 2, as can be seen in Figure 1, center-bottom. To better introduce the Token Shift technique, we define some notation. The **l**inear int**erp**olation (lerp) between $x_t$ and $x_{t-1}$ used in RWKV-4 and Eagle Token Shift is defined as:

$$\text{lerp}_\square(a, b) = a + (b - a) \odot \mu_\square \tag{3}$$

where each $\mu_\square \in \mathbb{R}^D$ is a learnable vector.

Token Shift allows the model to learn how much new versus old information should be allocated per time step to each channel of receptance, key, value, and gate vectors ($r, k, v,$ and $g$ respectively) independently and uniquely for each head. This makes it possible to form induction heads (Elhage et al., 2021) within a single layer since even a single head can directly accumulate both past and current token data into separate subspaces within these vectors.

**Eagle Time Mixing** The formula of Eagle Time Mixing can be written as follows:

$$\square_t = \text{lerp}_\square(x_t, x_{t-1})W_\square, \quad \square \in \{r, k, v, g\} \tag{4}$$

$$w = \exp(-\exp(\omega)) \tag{5}$$

$$\boldsymbol{wkv}_t = \text{diag}(u) \cdot k_t^{\text{T}} \cdot v_t + \sum_{i=1}^{t-1} \text{diag}(w)^{t-1-i} \cdot k_i^{\text{T}} \cdot v_i \in \mathbb{R}^{(D/h) \times (D/h)} \tag{6}$$

$$o_t = \text{concat}\left(\text{SiLU}(g_t) \odot \text{LayerNorm}(r_t \cdot \boldsymbol{wkv}_t)\right)W_o \in \mathbb{R}^D \tag{7}$$

Where LayerNorm operates on each of $h$ heads separately, which is also equivalent to the GroupNorm (Wu & He (2018)) operation on $h$ groups. It is also worth noting that $w$ is obtained from $w = \exp(-\exp(\omega))$, where $\omega \in \mathbb{R}^{D/h}$ are the actual headwise trainable parameters. This ensures that $w$ falls within the interval $(0, 1)$, guaranteeing that $\text{diag}(w)$ is a contraction matrix.

The $\boldsymbol{wkv}_t$ attention calculation can alternatively be written in a recurrent form:

$$\boldsymbol{wkv}' = \boldsymbol{s} + \text{diag}(u) \cdot k^{\text{T}} \cdot v \tag{8}$$

$$\boldsymbol{s}' = \text{diag}(w) \cdot \boldsymbol{s} + k^{\text{T}} \cdot v \tag{9}$$

RWKV's $\boldsymbol{wkv}$ term can be considered a decay-based equivalent to the normalised $k^{\text{T}}v$ term in Linear Attention. It is instructive to note how for a given head $j$ the recurrent state $s$ is a sum of $k^Tv$ where each channel of $s$ individually decays by the corresponding channel of $w$ at each time step. Prior to the application of the receptance vector, gating, and output weights, a per-channel learned boost $u$ is multiplied with the current token's $k^{\text{T}}v$ and summed with the state, as can be seen in Figure 1, top-right. This gives the current token special treatment relative to the sum of past tokens contained within the decaying state history. The receptance is multiplied by this sum, acting like the query term in Linear Attention.

**Channel Mixing** In both Eagle and Finch, the Channel Mixing module is identical to the previous RWKV-4 architecture, except for a slightly reduced hidden dimension from $4D$ to $3.5D$. This reduction accounts for new gating weights in Eagle Time Mixing to ensure an equi-parameter relation with the prior model at the same number of layers and embedding dimension. We do not further reduce the hidden dimension in Finch despite adding a small number of new parameters for LoRA weights. The formulas for Channel Mixing are the same as RWKV-4, but we restate them here to ensure notational consistency, using linear interpolation from Equation 3:

$$r'_t = \text{lerp}_{r'}(x'_t, x'_{t-1})W_{r'} \in \mathbb{R}^D \tag{10}$$

$$k'_t = \text{lerp}_{k'}(x'_t, x'_{t-1})W_{k'} \in \mathbb{R}^{3.5D} \tag{11}$$

$$v'_t = \text{ReLU}(k'_t)^2 W_{v'} \in \mathbb{R}^D \tag{12}$$

$$o'_t = \sigma(r'_t) \odot v'_t \in \mathbb{R}^D \tag{13}$$

## 4.2 Finch

**Finch Token Shift** The **d**ata-**d**ependent **l**inear int**erp**olation (ddlerp) between $x_t$ and $x_{t-1}$ used in Finch Token Shift is defined as:

$$\text{lora}_\square(x) = \lambda_\square + \tanh(xA_\square)B_\square \tag{14}$$

$$\text{ddlerp}_\square(a, b) = a + (b - a) \odot \text{lora}_\square(a + (b - a) \odot \mu_x) \tag{15}$$

where $\mu_x$ and each $\lambda_\square$ introduce a trainable vector of dimension $D$ and each $A_\square \in \mathbb{R}^{D \times 32}$, $B_\square \in \mathbb{R}^{32 \times D}$ introduce new trainable weight matrices. For the special case of LoRA$_\omega$ seen below we introduce double-sized trainable weight matrices $A_\omega \in \mathbb{R}^{D \times 64}$, $B_\omega \in \mathbb{R}^{64 \times D}$. A schematic representation can be found in Figure 1, bottom-right.

This new form of Token Shift enhanced with data-dependence is intended to expand the abilities of the model beyond the RWKV-4/Eagle style of Token Shift so that the amount of new and old data allocated per channel now depends on the input at both current and prior time steps.

**Finch Time Mixing**

$$\square_t = \text{ddlerp}_\square(x_t, x_{t-1})W_\square, \quad \square \in \{r, k, v, g\} \tag{16}$$

$$d_t = \text{lora}_d(\text{ddlerp}_d(x_t, x_{t-1})) \tag{17}$$

$$w_t = \exp(-\exp(d_t)) \tag{18}$$

$$\boldsymbol{wkv}_t = \text{diag}(u) \cdot k_t^{\mathrm{T}} \cdot v_t + \sum_{i=1}^{t-1} \text{diag}\left(\bigodot_{j=i+1}^{t-1} w_j\right) \cdot k_i^{\mathrm{T}} \cdot v_i \in \mathbb{R}^{(D/h) \times (D/h)} \tag{19}$$

$$o_t = \text{concat}\left(\text{SiLU}(g_t) \odot \text{LayerNorm}(r_t \cdot \boldsymbol{wkv}_t)\right) W_o \in \mathbb{R}^D \tag{20}$$

The $\boldsymbol{wkv}_t$ attention calculation can alternatively be written in a recurrent manner:

$$\boldsymbol{wkv}' = \boldsymbol{s} + \text{diag}(u) \cdot k^{\mathrm{T}} \cdot v \tag{21}$$

$$\boldsymbol{s}' = \text{diag}(w) \cdot \boldsymbol{s} + k^{\mathrm{T}} \cdot v \tag{22}$$

Unlike in Eagle, $w_t$ here is not static across the sequence (dashed arrows in Figure 1, left and top-right.). This is the core change to decay in Finch, as each channel of $w_t$ can now vary independently over time, in a data-dependent manner, whereas previously it was a fixed learned vector.

The new LoRA mechanisms above are used to take learned vectors, as seen in Eagle, and inexpensively augment them with additional offsets determined by the incoming input. Note that the LoRA process itself uses an Eagle style Token-Shifted value as its input, not just the latest token. The new time-varying decay $w_t$ goes one step further, applying LoRA again afterward. Intuitively, this is a second-order variant of Token-Shifting, allowing each channel of $w_t$ to vary based on a mix of the current and prior tokens, with the mix itself determined by aspects of both tokens.

## 5 Artifacts

We release our tokenizer, dataset, and model weights under an Apache 2.0 license

### 5.1 RWKV World Tokenizer

Tokenization is important in language modelling as it conditions the learning relationships between tokens and the generation of new text based on those patterns. The numbers of tokens to build a single semantic chunk are, however, often very unequally distributed against non-European and other underrepresented languages. Byte-pair-encoding (BPE) based tokenizers which are trained with this inequality result in not only lower performances against underrepresented languages but also undue economic costs such as inference Ahia et al. (2023) and continual pre-training with extended vocabulary Lin et al. (2024); Sasaki et al. (2023). To address these problems, we manually select tokens from multiple vocabulary files to well-represent non-European languages (Details in Appendix F). This tokenizer is implemented via a Trie (Prefix Tree) to boost speed while maintaining simplicity. Encoding is performed as matching the longest element in vocabulary with an input string from left to right. Our tokenizer's vocabulary construction aims to mitigate *undue* burden, which naive BPE and related methods cause on minority languages.

### 5.2 RWKV World v2 Dataset

We train our models on the new **RWKV World v2 Dataset**, a new multilingual 1.12 trillion token dataset drawn from a wide variety of hand selected publicly available data sources.

This dataset is designed to go beyond the English-heavy focus of many datasets widely used to train LLMs today. We do this to support usage by the majority of the worldwide population who are not native English speakers, to improve representation within model responses, and also to enable transfer learning so that our models can apply knowledge across cultures and locales. We put a strong emphasis on factual knowledge and code, but also on cultural works including stories, books, subtitles, and conversations. The source data is approximately 70% English, 15% multilingual, and 15% code. We describe the components of our dataset in detail in Appendix D.

### 5.3 Pre-Trained Models

We have pre-trained and publicly released the six Apache 2.0 licensed Eagle and Finch models: **Eagle 0.4B**, **Eagle 1.5B**, **Eagle 3B**, **Eagle 7B**, **Finch 1.6B**, and **Finch 3B**, all trained for 1 epoch on RWKV World v2 Dataset. See Appendix E for detailed parameter counts and FLOPs calculations.

## 6 Language Modeling Experiments

**LM Evaluation Harness Benchmarks** To assess the performance of Eagle and Finch models, we evaluate on a series of common multi-lingual and English-focused benchmarks using lm_evaluation_harness (Gao et al., 2023) as shown in Tables 3 and 4. Following the advice of Biderman et al. (2024), we reran evaluations of models that use a different evaluation framework in their papers.

Both Eagle and Finch demonstrate exceptionally high capabilities on multi-lingual benchmarks, with nearly all results significantly outperforming the other similarly sized models we tested.

In Appendix G.1, we plot the accuracy versus FLOPs used to train various open models across a similar set of common benchmarks, and find that Eagle and Finch provide competitive performance on these multilingual benchmarks. The two models additionally obtain competitive performance across these English benchmarks. See figures 4 and 5.

| Model | lmb.m ppl ↓ | lmb.m acc ↑ | pawsx acc ↑ | xcopa acc ↑ | xnli acc ↑ | xsClz acc ↑ | xwin acc ↑ | avg acc ↑ |
|---|---|---|---|---|---|---|---|---|
| Pythia-1.4b | 115.9 | 35.5 | 50.9 | 52.7 | 38.9 | 51.8 | 68.3 | 49.7 |
| Mamba-1.4b | 73.1 | 40.4 | 48.0 | 54.4 | **41.6** | 54.2 | 72.4 | 51.8 |
| RWKV-4-1.5b | 72.5 | 38.5 | **53.7** | 55.4 | 39.3 | 56.0 | 67.7 | 51.8 |
| Eagle-1.5b | 43.2 | 44.8 | 51.9 | 57.9 | 40.4 | **57.9** | 73.0 | 54.3 |
| **Finch-1.6b** | **37.5** | **46.9** | 50.9 | **58.0** | 41.4 | **57.9** | **74.9** | **55.0** |
| Pythia-2.8b | 81.3 | 38.8 | 49.4 | 53.7 | 40.0 | 53.5 | 71.5 | 51.1 |
| Mamba-2.8b | 53.7 | 43.5 | 43.6 | 55.3 | 42.1 | 56.3 | 75.6 | 52.7 |
| RWKV-4-3b | 48.1 | 43.4 | 50.9 | 57.5 | 40.9 | 58.1 | 72.3 | 53.9 |
| Eagle-3b | 30.8 | 49.1 | **51.6** | 59.0 | 42.3 | 59.8 | 76.9 | 56.5 |
| **Finch-3b** | **28.1** | **50.5** | 49.7 | **59.5** | **44.2** | **60.7** | **77.8** | **57.1** |
| Pythia-6.9b | 85.6 | 36.7 | 48.4 | 54.1 | 40.0 | 54.2 | 70.9 | 50.7 |
| MPT-7b | 49.8 | 44.4 | 43.5 | 53.6 | 39.8 | 56.3 | 76.9 | 52.4 |
| Llama-2-7b | 30.4 | 50.8 | 41.2 | 56.7 | 39.9 | 57.5 | 79.5 | 54.3 |
| Falcon-7b | 28.7 | 51.3 | 48.2 | 56.0 | 39.0 | 56.0 | 77.7 | 54.7 |
| Mistral-7B-v0.1 | 27.1 | 51.9 | 41.5 | 55.9 | 43.1 | 59.2 | **81.2** | 55.5 |
| RWKV-4-7b | 33.1 | 47.4 | **52.1** | 60.1 | 41.2 | 60.9 | 76.5 | 56.4 |
| **Eagle-7B** | **21.0** | **53.7** | 45.6 | **62.2** | **44.0** | **63.3** | 80.4 | **58.2** |

Table 3: Multilingual Benchmarks, including LAMBADA Multilingual (**lmb.m**) (Gao et al., 2023), XCOPA (Ponti et al., 2020), XNLI (Conneau et al., 2018), PAWS-X (Yang et al., 2019), XStoryCloze (**xsClz**) (Lin et al., 2022), xWinograd (**xwin**) (Tikhonov & Ryabinin, 2021).

**Associative Recall** Associative recall (AR) is a synthetic task designed to mimic the way humans associate and retrieve information. It measures a model's proficiency in recalling

| Model | lmb.o acc ↑ | hella acc_n ↑ | piqa acc ↑ | arcE acc ↑ | arcC acc ↑ | glue acc ↑ | winG acc ↑ | sciq acc ↑ | copa acc ↑ | avg acc ↑ |
|---|---|---|---|---|---|---|---|---|---|---|
| Pythia-1.4b | 61.0 | 52.0 | 70.8 | 61.4 | 26.2 | 47.1 | 57.3 | 86.5 | 71.0 | 59.2 |
| RWKV-4-1.5b | 60.1 | 51.6 | 71.5 | 58.4 | 27.1 | 46.1 | 55.2 | 84.7 | 78.0 | 59.2 |
| Eagle-1.5b | 65.7 | 55.0 | 71.1 | 62.2 | 28.7 | **54.1** | 59.1 | **89.7** | 76.0 | 62.4 |
| **Finch-1.6b** | **66.8** | 57.3 | 72.6 | 62.7 | 29.8 | 49.8 | 59.4 | 89.6 | 78.0 | **62.9** |
| **Mamba-1.4b** | 64.5 | **59.0** | **74.2** | 65.0 | **30.1** | 47.0 | **61.3** | 87.1 | **80.0** | 63.1 |
| Pythia-2.8b | 63.8 | 59.1 | 73.9 | 63.8 | 29.0 | 47.3 | 58.2 | 88.6 | 79.0 | 62.5 |
| RWKV-4-3b | 65.7 | 58.8 | 72.4 | 62.9 | 32.4 | 53.6 | 57.5 | 87.6 | **86.0** | 64.1 |
| Eagle-3b | 68.7 | 62.6 | 74.3 | 68.6 | 33.8 | 46.3 | 62.0 | **92.6** | 85.0 | 66.0 |
| Mamba-2.8b | 68.1 | **65.9** | **75.2** | **69.7** | 33.8 | 46.3 | 63.0 | 90.2 | 84.0 | 66.2 |
| **Finch-3b** | **70.8** | 64.8 | 74.2 | 66.5 | **34.6** | **58.2** | **63.6** | 92.5 | 82.0 | **67.5** |
| Pythia-6.9b | 60.9 | 63.2 | 74.8 | 66.5 | 32.0 | 47.7 | 61.5 | 88.9 | 79.0 | 63.8 |
| RWKV-4-7b | 69.8 | 65.3 | 75.0 | 67.4 | 34.0 | 56.4 | 62.4 | 90.8 | 85.0 | 67.3 |
| MPT-7b | 68.7 | 76.3 | 79.3 | 74.9 | 39.7 | 48.7 | 68.1 | 93.9 | 88.0 | 70.9 |
| Llama-2-7b | 73.5 | 76.0 | 78.1 | 76.4 | 43.1 | 42.9 | 69.1 | 93.9 | 87.0 | 71.1 |
| Falcon-7b | 74.6 | 76.4 | 79.5 | 74.8 | 40.3 | 45.8 | 67.1 | 94.4 | 88.0 | 71.2 |
| Eagle-7B | 74.2 | 70.9 | 77.0 | 73.8 | 39.5 | **57.5** | 67.4 | 95.5 | 88.0 | 71.5 |
| **Mistral-7B-v0.1** | **75.5** | **81.0** | **80.5** | **80.8** | **50.1** | 51.5 | **73.6** | **95.9** | **93.0** | **75.8** |

Table 4: English Focused Benchmarks, including LAMBADA (**lmb.o**) (Paperno et al., 2016), Hellswag (**hella**) (Hampel, 1974), PIQA (Bisk et al., 2020), AI2 ARC (**arcE**, **arcC**) (Bhakthavatsalam et al., 2021), GLUE (Wang et al., 2018), Winogrande (**winG**) (Sakaguchi et al., 2021), SciQ (Welbl et al., 2017), COPA (Roemmele et al., 2011).

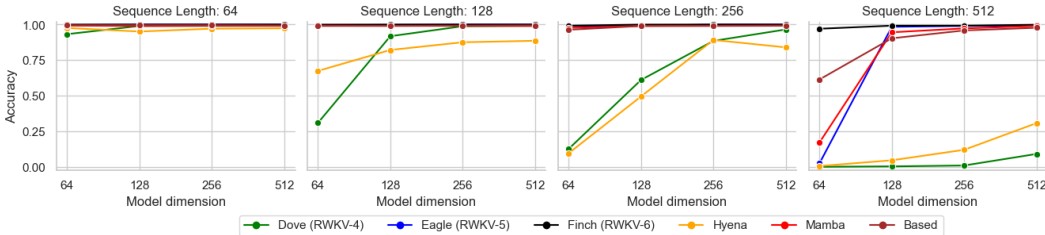

Figure 2: MQAR tasks. An increase in sequence length correlates with increased task difficulty.

information previously mentioned in context. Prior research suggests that a model's ability to perform AR is an indicative of its effectiveness in in-context learning (Elhage et al., 2021; Olsson et al., 2022). As a result, AR has been adopted as a benchmark in designing new language model architectures. (Fu et al., 2023; Poli et al., 2023; Lutati et al., 2023). Arora et al. (2023) benchmarked a range of models for multi-query associative recall (MQAR) and identified a performance gap between various linear transformer architectures and the transformer with attention. In MQAR tasks, prior RWKV models demonstrated a correlation between model dimension and sequence length. To compare architectures, we trained models using RWKV-4, Eagle and Finch on MQAR, using identical criteria with various model dimensions and sequence lengths. Eagle and finch show significant improvements in MQAR, notably, Finch achieves extremely high accuracy in MQAR in our tests, and outperforms all previously well-known non-transformer architectures for large language models. Our experiments reveal performance disparities between Mamba (Gu & Dao, 2023) and Finch, despite their shared architectural features such as matrix-valued state and data-dependent memory modification, suggesting different combinations of these elements result in superior performance.

**Long Context Experiments** We test loss versus sequence position on PG19 (Rae et al., 2019) test set of books from token 2048 onward across RWKV-4, Eagle and Finch. Despite having been trained solely on sequence length 4096, Eagle improves dramatically over RWKV-4

on this long sequence task, and Finch further improves on this test beyond Eagle, with loss continuing to drop further into the sequence. See Figure 6 for details.

## 7 Multimodal Experiments

**RWKV Music Modelling** RWKV is also capable of modelling musical score sequences. To compare the improvement in music modelling of RWKV-5 architecture over previous models, we use the Irishman ABC music dataset (Wu et al., 2023) and train a RWKV-5-Music model using the same hyperparameters as the existing RWKV-4-Music model. The model has $L = 25$ layers in total, with dimension $D = 512$ and a byte-level tokenizer of $V = 128$ tokens. The training context length is 1024. We use all 2162 pieces in the validation set and calculate the loss for each position from start. Loss is averaged across all pieces of music then Gaussian smoothed over positions in sequence, as drawn in Figure 8. The first 30-100 bytes of the ABC format are the file header and control codes, followed by the musical scores. RWKV-5 achieved approximately 2% loss improvement over RWKV-4, shown mainly in the musical score part, signifying that RWKV-5 has stronger modelling and generalization capabilities than its predecessor.

**VisualRWKV** VisualRWKV is the visual-enhanced version of RWKV language model for vision tasks. It follows a similar architecture to popular vision-language models (Liu et al., 2023a), consisting of a vision encoder and a language model. The architecture is shown in Figure 9. Specifically, we use LLaVA-1.5 as the dataset (Liu et al., 2023a), CLIP (Radford et al., 2021) as the vision encoder and Eagle 1.5B and 3B as the language model. VisualRWKV is trained with a two-stage instruction-tuning process to enhance model performance. The initial stage is pre-training for feature alignment, during which only the projection layer is subjected to updates. Next, we move on to the fine-tuning end-to-end stage, where both the projection layer and the RWKV language model are fine-tuned, but the vision encoder is still kept frozen. Table 15 shows that VisualRWKV is powerful for both visual understanding and reasoning. With a smaller vision encoder CLIP-L (0.4B) and modest-sized LLMs of 1.5B and 3B, it achieves results comparable to the combination of CLIP-G (1.0B) and CLIP-H (1.0B) with larger LLMs of 7B and 13B, and even outperforms larger models in some benchmarks.

## 8 Conclusions

We introduced Eagle (RWKV-5) and Finch (RWKV-6), marking substantial progress in RNN-based language modelling by integrating multiheaded matrix-valued states and dynamic data-driven recurrence mechanisms. These models demonstrate exceptional performance on MQAR and diverse linguistic benchmarks, rivaling Transformer architectures while retaining key RNN advantages. With models trained on a diverse multilingual corpus and available under Apache 2.0 license, our work not only advances the capabilities of language models but also fosters community accessibility and applicability across domains. While acknowledging the computational and ethical challenges, we hope that Eagle and Finch's efficient new architecture and wide availability will help push the boundaries of language modeling and pave the way for future innovations.

**Limitations and Future Work** We experimented with using Eagle as an embedding model on the Massive Text Embedding Benchmark (MTEB) (Muennighoff et al., 2023) but were unable to get strong performance. We believe that its state is a very high-quality embedding of the context but an appropriate method is required to aggregate the information content. We leave this to future work.

Our 1.12 trillion token corpus is much smaller than the training data sizes for contemporary models such as LLaMA2 (Touvron et al., 2023). Expanding our training corpus to be more diverse and expansive is a key priority to improving model performance (Albalak et al., 2024), though its also possible that this can be achieved by training for multiple epochs (Muennighoff et al., 2024). We also plan to train and release larger versions of Finch such as 7B and 14B parameters, and further extend its performance with reduced costs via Mixture of Experts (Shazeer et al., 2017).

## Acknowledgments

We thank Stability AI for the compute used to train our models and for technical support in the development of RWKV. We also thank the members of the RWKV and EleutherAI Discord servers for their help and work on further extending the applicability of RWKV to different domains. We also thank Shenzhen Yuanshi Intelligence Co., Ltd. for its contribution to the promotion and commercialization of RWKV. We thank Songlin Yang for assistance with the code and ideas for our time-parallel implementations.

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

# A Author Contributions

**Bo Peng** Original RWKV-5 and RWKV-6 ideas, original code, performance optimizations, original experiments, tokenizer design, dataset composition, and trained models from 0.4B to 7B.

**Daniel Goldstein** RWKV-5 and RWKV-6 time-parallelization research and code. Manuscript organization, initial draft sections 2, 3, 4, 5.1, 5.2, 6, G.2, and appendices B, D, N, and O. Proofreading and revisions of full manuscript. Experiments for 6 and G.2. Additional work on tables 1, 2, figure L, and appendix J.

**Quentin Anthony** Led manuscript and results organization. Revisions and proofreading of manuscript.

**Alon Albalak** Manuscript organization, initial draft of section 1, proofreading, formatting, and revisions of full manuscript.

**Eric Alcaide** Figure 1. Proofreading, formatting, and revisions of full manuscript.

**Stella Biderman** Oversight and planning on scaling figures and FLOP results. Manuscript assistance.

**Eugene Cheah** Experiments for section 6.

**Teddy Ferdinan** Self-Learning Capability (SLC) evaluation (Sec. G.6) – implementation of the method, performing experiments, initial draft of the section, description of the results (Tab. 13).

**Przemysław Kazienko** Planning the experiment with Self-Learning Capability (SLC) evaluation (Sec. G.6), supervising SLC experiments.

**Kranthi Kiran GV** Manuscript (sections 6 and 7; revision and proofreading). Tables 3 and 4.

**Haowen Hou** VisualRWKV based on RWKV-5, which encompasses original code, original experiments for Table 15, and trained models ranging from 1.5 billion to 3 billion parameters. Figure 9 and draft section 7. Proofreading and formatting fixes.

**Satyapriya Krishna** Primarily contributed to the evaluations of the models (Section **??** and **??**), and also made edits/improvements throughout the document.

**Ronald McClelland Jr.** Tables 1 and 2. Dataset research. Proofreading and formatting fixes.

**Niklas Muennighoff** Investigated using the RWKV models for embedding.

**Fares Obeid** RWKV-5 and RWKV-6 time-parallelization research. Section L. Experiments for figures 10 and 11. Proofreading full manuscript.

**Atsushi Saito** Section 1, 5.1, 6 and 6. Experiments for 6. Proofreading and adding citations.

**Guangyu Song** Section 6. Initial draft sections 1, 8. Experiments for 6. Contributions to table 1. Proofreading and citations.

**Haoqin Tu** Section G.5, experiments for Table 12. Proofreading full manuscript.

**Ruichong Zhang**   Initial paper structure organization, draft sections 3, 4, 5.1, 5.3 and appendices E, F, J and K. Experiments for music of section 7 and alignment of section G.3. Figure 8 and 3. Additional work on section 8 and appendix B. Proofreading and revision.

**Bingchen Zhao**   Section G.4, experiments for Figure 7. Proofreading full manuscript.

**Qihang Zhao**   Section 2, Tables 1. Proofreading and revisions.

**Jian Zhu**   Initial draft sections 2 and C. Captions of Table 4, 3 and 6. Fixing citations and formatting the whole manuscript. Proofreading and revisions.

**Rui-Jie Zhu**   Optimizing draft Section C, reorganizing Table 6, 13, and 11. Proofreading and revisions.

# B   Additional Architecture Details

The **WKV** computations of Eagle and Finch can be parallelized across the time dimension using a variety of techniques including associative scan or the parallelization techniques used in FlashAttention. (Dao et al., 2022) The simplest of these, while highly parallel, prove inefficient due to repeated expensive memory transfers between fast SRAM and slower HBM. We take a different approach when training, choosing to parallelize over non-time dimensions only while using a custom CUDA implementation that carefully keeps state operations in fast SRAM, which is simpler yet provides enough breadth for a highly efficient implementation. See Appendix L for kernel experiments. We provide an additional pure PyTorch implementation with similar full-model speed characteristics that parallelizes over the time dimension using an algorithmic approach similar to GLA (Yang et al., 2023).

Unlike Transformers, RWKV's recurrence mechanism does not examine tokens more than one time-step old. This allows us to train on and provide inference for unbounded sequence lengths without requiring increased computing power or memory. Another significant advantage is that RWKV does not utilize explicit positional encoding, which allows RWKV to handle contexts of arbitrary length without modification.

**Finch Token Shift**   Finch changes the token shift mechanism to become data-dependent. Intuitively, important information can effectively flag itself for inclusion using this mechanism, and less important information can flag itself to partially or fully avoid entering the data stream, leaving room for more important pre-existing data to remain. Viewed from the perspective of induction heads, we theorize that this could allow for potential misleading matches to be pre-filtered out up front if they are not deemed useful for a given task.

**Improved WKV (Weighted Key-Value State) Modules**   The Eagle WKV attention sub-module is similar to the linear attention mechanism found in RetNet, but with learned per-channel decay rates replacing RetNet's static per-head decay rates. Our matrix-valued states feature a geometrically decaying $K^{\mathrm{T}}V \in \mathbb{R}^{(D/h)\times(D/h)}$ term. This term can be intuitively understood as a memory bank of values, with $K$ acting as an input gate for rows receiving the current token embedding's value. Each row of this state decays at its own rate via the learned parameter $w$.

In Finch, we augment the learned token-shift parameters $\mu_r, \mu_k, \mu_v, \mu_w$ and decay rate parameter $w$ with learned weight matrices. Inspired by Low-Rank Adaptation (LoRA) (Hu et al., 2022), we provide two new learned weight matrices for each such parameter $y$, computing $y' = y + \tanh(xA)B$. This approach allows us to dynamically generate data-dependent token-shift amounts and decay rates with only modest increases in computational cost and model size.

**Extra SiLU Gating**   We remove the Sigmoid activation of receptance in favor of a new SiLU gate on the output of our linear attention calculation. Our receptance term now functions much like the query term in linear attention.

**Eagle and Finch Linear Attention Formula, PyTorch Recurrent Implementation**  .

```
1  # r, k, v parameter shape (B,H,1,D//H)
2  # w parameter of shape (1,H,1,D//H) for Eagle (RWKV-5),
3  #                      (B,H,1,D//H) for Finch (RWKV-6)
4  # u parameter of shape (1,H,1,D//H)
5  # wkv_state parameter of shape (B,H,D//H,D//H)
6  def rwkv_5_or_6_recurrent(r, k, v, w, u, wkv_state):
7      kv = k.mT @ v
8      out = r @ (wkv_state + u.mT * kv)
9      wkv_state = w.mT * wkv_state + kv
10     return out, wkv_state
```

**Evolution of RWKV Formula in Expanded form**   Table 5 shows the expansion of terms at each sequence position to illustrate the progression of changes from RWKV-4 through RWKV-6. The main change from RWKV-4 to RWKV-5 is the elimination of denominator and incorporation of matrix states. RWKV-6 introduces the sequential dependence of $w$ which becomes $w_t$.

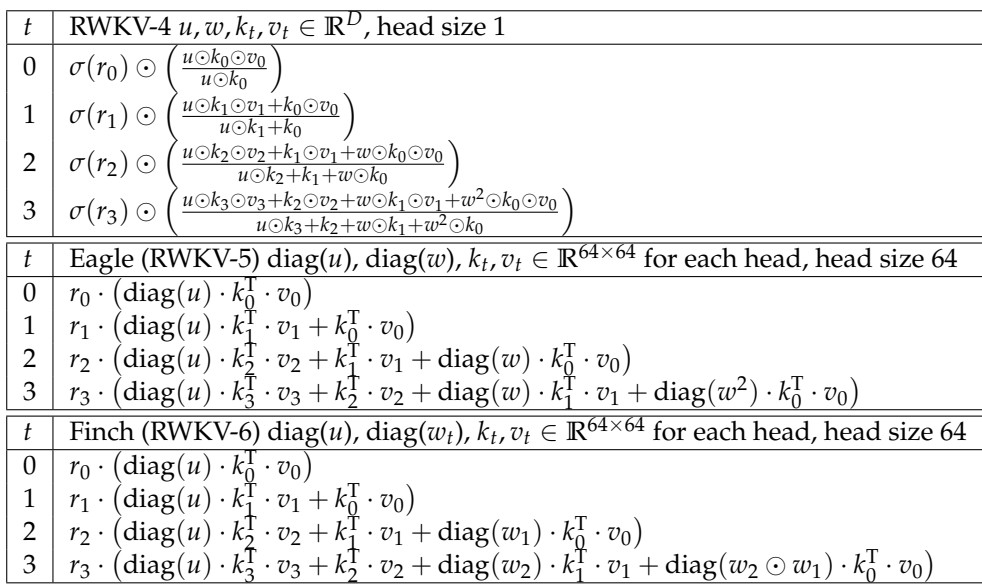

| $t$ | RWKV-4 $u, w, k_t, v_t \in \mathbb{R}^D$, head size 1 |
|---|---|
| 0 | $\sigma(r_0) \odot \left( \frac{u \odot k_0 \odot v_0}{u \odot k_0} \right)$ |
| 1 | $\sigma(r_1) \odot \left( \frac{u \odot k_1 \odot v_1 + k_0 \odot v_0}{u \odot k_1 + k_0} \right)$ |
| 2 | $\sigma(r_2) \odot \left( \frac{u \odot k_2 \odot v_2 + k_1 \odot v_1 + w \odot k_0 \odot v_0}{u \odot k_2 + k_1 + w \odot k_0} \right)$ |
| 3 | $\sigma(r_3) \odot \left( \frac{u \odot k_3 \odot v_3 + k_2 \odot v_2 + w \odot k_1 \odot v_1 + w^2 \odot k_0 \odot v_0}{u \odot k_3 + k_2 + w \odot k_1 + w^2 \odot k_0} \right)$ |

| $t$ | Eagle (RWKV-5) $\mathrm{diag}(u), \mathrm{diag}(w), k_t, v_t \in \mathbb{R}^{64 \times 64}$ for each head, head size 64 |
|---|---|
| 0 | $r_0 \cdot \left( \mathrm{diag}(u) \cdot k_0^\mathsf{T} \cdot v_0 \right)$ |
| 1 | $r_1 \cdot \left( \mathrm{diag}(u) \cdot k_1^\mathsf{T} \cdot v_1 + k_0^\mathsf{T} \cdot v_0 \right)$ |
| 2 | $r_2 \cdot \left( \mathrm{diag}(u) \cdot k_2^\mathsf{T} \cdot v_2 + k_1^\mathsf{T} \cdot v_1 + \mathrm{diag}(w) \cdot k_0^\mathsf{T} \cdot v_0 \right)$ |
| 3 | $r_3 \cdot \left( \mathrm{diag}(u) \cdot k_3^\mathsf{T} \cdot v_3 + k_2^\mathsf{T} \cdot v_2 + \mathrm{diag}(w) \cdot k_1^\mathsf{T} \cdot v_1 + \mathrm{diag}(w^2) \cdot k_0^\mathsf{T} \cdot v_0 \right)$ |

| $t$ | Finch (RWKV-6) $\mathrm{diag}(u), \mathrm{diag}(w_t), k_t, v_t \in \mathbb{R}^{64 \times 64}$ for each head, head size 64 |
|---|---|
| 0 | $r_0 \cdot \left( \mathrm{diag}(u) \cdot k_0^\mathsf{T} \cdot v_0 \right)$ |
| 1 | $r_1 \cdot \left( \mathrm{diag}(u) \cdot k_1^\mathsf{T} \cdot v_1 + k_0^\mathsf{T} \cdot v_0 \right)$ |
| 2 | $r_2 \cdot \left( \mathrm{diag}(u) \cdot k_2^\mathsf{T} \cdot v_2 + k_1^\mathsf{T} \cdot v_1 + \mathrm{diag}(w_1) \cdot k_0^\mathsf{T} \cdot v_0 \right)$ |
| 3 | $r_3 \cdot \left( \mathrm{diag}(u) \cdot k_3^\mathsf{T} \cdot v_3 + k_2^\mathsf{T} \cdot v_2 + \mathrm{diag}(w_2) \cdot k_1^\mathsf{T} \cdot v_1 + \mathrm{diag}(w_2 \odot w_1) \cdot k_0^\mathsf{T} \cdot v_0 \right)$ |

Table 5: Evolution of the RWKV Formula

## C  Additional Related Work

**Efficient transformers**   Recently there have been many attempts to improve upon the original transformer time complexity and memory usage, while maintaining or improving performance. Many of these efficient transformer variants use some form of nonuniform or local attention mechanisms or a combination thereof. For example, LongFormer (Beltagy et al., 2020) makes use of the sliding window attention and BigBird (Zaheer et al., 2020) adopts randomized sparse and random attention patterns to approximate full attention. Similar examples also include LongT5 (Guo et al., 2022) and StreamingLLM (Xiao et al., 2023). Instead of using fixed patterns, Reformer (Kitaev et al., 2019) and Sparse Sinkhorn attention (Tay et al., 2020) learn to dynamically pay attention to selected tokens. Variants including Linformer (Wang et al., 2020), Nyströmformer (Xiong et al., 2021) and Performer (Choromanski et al., 2020) apply matrix approximation methods to approximate the full attention matrix but with lower computational complexity.

The Attention Free Transformer (AFT) (Zhai et al., 2021) introduces a modified form of linear attention (Katharopoulos et al., 2020a) , where the number of attention heads is equal

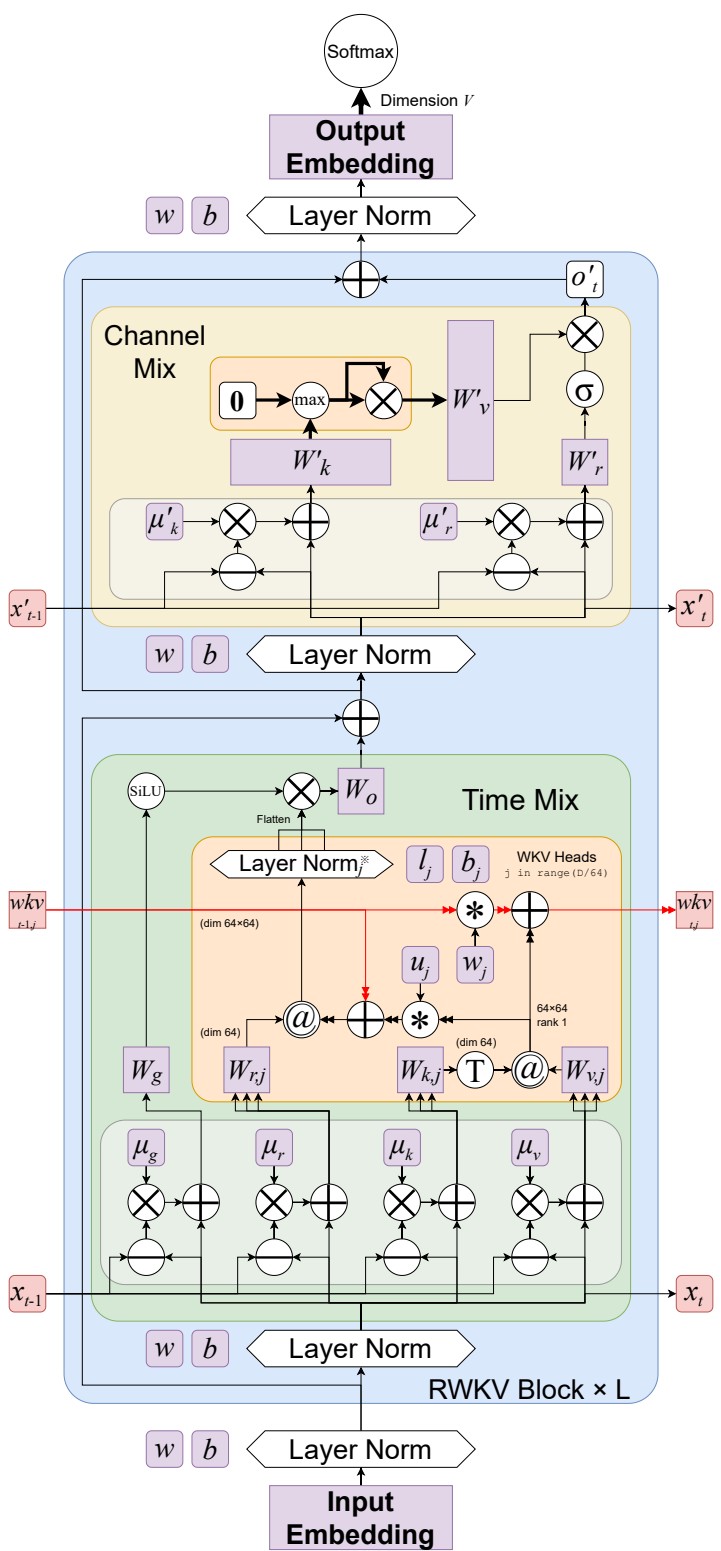

Figure 3: Eagle Overall Architecture.

to the size of the feature dimension. It also incorporates a set of learned pairwise positional biases, denoted as $w$. The AFT can be conceptualized as calculating a per-channel weighted average of values. The weight for a specific location is determined by the sum of the key at that location and the corresponding learned positional bias.

Token-shift, as first seen in RWKV-4, is a learned per-channel linear interpolation between the current input and the input at the previous time step, intended to enhance the model with a computationally inexpensive mechanism for choosing between new versus older information within various embedding sub-spaces and for forming induction heads even within a single layer. It is instructive to compare token-shift to a 1D convolution with kernel length 2, as it operates in a similar manner but reuses its parameters via an enforced linear relationship. Recent SSMs have begun using short convolutions in a similar placement within their architectures, typically with kernel length 3 to 4. (Poli et al., 2023; Gu & Dao, 2023)

Retentive Networks (RetNet) (Sun et al., 2023) introduces a fixed decay rate schedule and xPos (Sun et al., 2022) to linear attention. This design combines positional information with an inductive bias towards recency while still allowing both RNN and parallel implementations.

Please refer to Tay et al. (2022) and Wan et al. (2023) for a comprehensive and in-depth survey of efficient transformers.

**Recurrent architectures**   Before the advent of transformers, recurrent neural networks, especially Long Short-Term Memory (LSTM) (Hochreiter & Schmidhuber, 1997) and Gated Recurrent Unit (GRU) (Cho et al., 2014), were the dominant architectures in NLP for sequence processing. However, traditional RNNs are hard, if not impossible, to parallelize across the time dimension, susceptible to gradient vanishing and explosion, and ineffective in capturing long-range dependencies, which are ubiquitous in natural language. These shortcomings contributed to the rapid decline of traditional RNNs in NLP.

There has been a revival of RNNs in NLP research (Tiezzi et al., 2024) in recent years. Compared to transformers with quadratic complexity, RNNs are highly efficient in autoregressive inference with O(1) time complexity per step, making them an attractive architecture for large language models. Many efforts have been devoted to parallelized recurrent models and improving their capability to capture long-range dependency, while maintaining the low inference complexity.

The Legendre Memory Unit (LMU) (Voelker et al., 2019) was designed to efficiently handle long-range dependencies with a new type of memory cell for recurrent neural networks. Unlike LSTM units, which struggle with remembering information over very long sequences, LMU use Legendre polynomials to create a memory system that can maintain and process information over extended time periods more effectively. High-order polynomial projection operators (HiPPO) (Gu et al., 2020) generalizes LMU by providing a flexible framework for online compression of signals through polynomial projections, accommodating various polynomial bases beyond Legendre polynomials. It optimizes function approximation over time, adapting to different data timescales without needing predefined hyperparameters. SSMs have inspired a range of follow-up research to incorporate SSMs, or modified SSMs into end-to-end architectures for language modeling, including MEGA (Ma et al., 2022), DSS (Gupta et al., 2022), H3 (Fu et al., 2022), and Linear Recurrent Unit (LRU) (Orvieto et al., 2023).

Mamba (Gu & Dao, 2023) is a selective SSM that introduces time-dependent selective mechanism to enhance the long-range modeling ability of SSMs. The selectivity removes the linear time-variance property of the SSM, making it no longer possible to parallelize Mamba as a long convolution kernel. Yet Mamba can still be effectively parallelized using parallel associative scan (Blelloch, 1990; Martin & Cundy, 2018; Smith et al., 2022) with a hardware-aware implementation. Recently proposed GateLoop (Katsch, 2023) also adopts a similar data-dependent state transitions. The data-dependent states, also concurrently proposed in GLA (Yang et al., 2023), are similar to the Weighted Key-Value State in Finch.

| Dataset | Domain | Dataset | Domain |
|---|---|---|---|
| Wikipedia[a] | Encyclopedia | marianna13/vault_text | Books |
| SlimPajama | Web | marianna13/random_quora | Forums |
| peS2o | Academia | marianna13/zlib | Books |
| BigPatent | Patents | minipile | Various |
| Pile of Law | Legal, Administrative | tatoeba | Multilingual Translations |
| StarCoder[b] | Code | poetry-foundation | Poetry |
| OSCAR23.01[c] | Multilingual Web | proof-pile | Academia: Math |
| TED2020 | Transcripts: TED, TEDx | reddit-math | Forums: Math |
| PhilPapers | Academia: Philosophy | soda | Dialogue |
| NIH-ExPORTER | Grants: NIH | song_lyrics | Lyrics |
| EuroParl | Multilingual Legal | TinyStories | Stories |
| Enron-Emails | Emails | walkthroughs2020 | Game Walkthroughs |
| Ubuntu IRC | Chat | wikihow-qa-16k | How-To |
| HackerNews | Forums | Alpaca | Various |
| OpenWebText2 | Web | camel-ai/math | Math |
| Gutenberg PG-19 | Books | camel-ai/code | Code |
| Books3 | Books | camel-ai/physics | Physics |
| OpenSubtitles | Subtitles | camel-ai/chemistry | Chemistry |
| YTSubtitles | Subtitles | camel-ai/ai_society | Job Roles |
| ao3_skylion | Stories | camel-ai/biology | Biology |
| honeyfeed-3600 | Stories | Dolly | Various |
| scribble-17k | Stories | Evol-Instruct | Various |
| syosetu711k | Stories (Japanese) | gpt4all | Code |
| marianna13/fanfics | Stories | Guanaco | Various Multilingual |
| marianna13/gamedev | Forums | LaMini | Various |
| marianna13/ia-books | Books | oasst1 | Multilingual Conversations |
| marianna13/libgen | Textbooks, Books | ShareGPT | Conversations |
| marianna13/research_gate | Academia | UltraChat | Conversations |
| marianna13/superuser | Forums | BELLE 10M Chinese | Various Chinese |
| marianna13/the-eye | Books | | |

Table 6: Components of the RWKV World v2 dataset, their source links, and their domains.
[a]For Wikipedia, we include all languages from date 04/01/2023, with certain overrepresented languages randomly subsampled (see wiki.txt in the supplementary material for exact amounts)
[b]For StarCoder, we included only those datasets with at least 10 stars
[c]For OSCAR23.01, we include non-English languages only, with certain languages randomly subsampled (see oscar.txt in the supplementary material for exact amounts)

A contemporary but independent work also proposes recurrent models named as Hawk and Griffin (De et al., 2024). Hawk is a recurrent model with the Real-Gated Linear Recurrent Unit (RG-LRU), whereas Griffin mixes the RG-LRU with local multi-query attention, thereby achieving long-context extrapolation efficiently.

Please see Tiezzi et al. (2024) and Cirone et al. (2024) for a comprehensive review of recent developments of recurrent models.

## D Training Dataset Details

Most of the component data sources for the RWKV World v2 dataset are used intact, with no up- or down-sampling done so all tokens are given equal weighting. Recent works have demonstrated the impact that automated data mixing can have on pretraining (Albalak et al., 2023; Xie et al., 2024), but we leave this as an exploration for future work. Some sub-sampling is done for over-represented languages within a few data sources. All tokens are given equal weighting unless otherwise noted in Table 6.

| SlimPajama | Soboleva et al. (2023) |
|---|---|
| StarCoder | Li et al. (2023b) |
| OSCAR23.01 | Suárez et al. (2019) |
| TED2020 | Reimers & Gurevych (2020) |
| the Pile | Gao et al. (2020) |
| Evol-Instruct | Xu et al. (2023) |

Table 7: RWKV World v2 dataset component citations

# E  Computing Costs

| Model Name | $L$ | $D$ | State Size | Parameters | InferFLOPs | TrainFLOPs |
|---|---|---|---|---|---|---|
| Eagle 0.4B | 24 | 1024 | 1 622 016 | $4.62 \times 10^8$ | $9.33 \times 10^8$ | $2.80 \times 10^9$ |
| Eagle 1.5B | 24 | 2048 | 3 244 032 | $1.58 \times 10^9$ | $3.17 \times 10^9$ | $9.52 \times 10^9$ |
| Eagle 3B | 32 | 2560 | 5 406 720 | $3.06 \times 10^9$ | $6.16 \times 10^9$ | $1.85 \times 10^{10}$ |
| Eagle 7B | 32 | 4096 | 8 650 752 | $7.52 \times 10^9$ | $1.51 \times 10^{10}$ | $4.53 \times 10^{10}$ |
| Finch 1.6B | 24 | 2048 | 3 244 032 | $1.60 \times 10^9$ | $3.22 \times 10^9$ | $9.66 \times 10^9$ |
| Finch 3B | 32 | 2560 | 5 406 720 | $3.10 \times 10^9$ | $6.23 \times 10^9$ | $1.87 \times 10^{10}$ |

Table 8: Released Eagle and Finch model details and FLOP counts. Inference and training FLOPs are per token numbers.

Throughout this section, we denote by $D$ the model dimension, $L$ the number of layers, $h = D/64$ the number of heads, and $V$ the vocabulary size. All models are trained with $V = 65536$.

The number of parameters for all Eagle models is computed by the formula:

$$\#(\text{Params})_{\text{E}} = 13D^2L + 14DL + 4D + 2DV \tag{23}$$

The FLOPs for inference is one forward pass for each token. It is approximated by twice the number of parameters (for matrices, there is one addition and one multiplication for each entry) plus six times the size of **WKV** internal states (see 7 8 9), which is

$$\#(\text{InferFLOPs})_{\text{E}} = 2(13D^2L + 14DL + 4D + 2DV) + 6D^2L/h \tag{24}$$
$$= 26D^2L + 28DL + 8D + 4DV + 6D^2L/h \tag{25}$$

The FLOPs for training are approximated as three times the FLOPs of the forward pass without the last term, yielding a total FLOPs of

$$\#(\text{TrainFLOPs})_{\text{E}} = 78D^2L + 84DL + 16D + 12DV + 18D^2L/h \tag{26}$$

These numbers for Finch are marginally larger:

$$\#(\text{Params})_{\text{F}} = 13D^2L + 464DL + 4D + 2DV \tag{27}$$
$$\#(\text{InferFLOPs})_{\text{F}} = 26D^2L + 928DL + 8D + 4DV + 6D^2L/h \tag{28}$$
$$\#(\text{TrainFLOPs})_{\text{F}} = 78D^2L + 2784DL + 24D + 12DV + 18D^2L/h \tag{29}$$

In both Eagle and Finch, one needs an internal state to store some previous information, just like any other RNN. In each layer, the internal state consists of three parts:

1. The most recent single-timestep input to the Time-mixing module, denoted as $x_{t-1} \in \mathbb{R}^D$, useful for the Token Shift.
2. The most recent single-timestep input to the Channel-mixing module, denoted as $x'_{t-1} \in \mathbb{R}^D$, also useful in Token Shift.

3. WKV head memory: Denoted by $\boldsymbol{wkv}_{t,j} \in \mathbb{R}^{(D/h) \times (D/h)}$, for $j = 1, 2, \cdots, h$. This is the core part of the internal state that dominates the most information.

The total size of the Eagle and Finch internal state is

$$\#(\text{State}) = L(2D + D^2/h) = 66DL \tag{30}$$

It's worth noting that the internal state size of Eagle and Finch is more than an order of magnitude bigger than RWKV-4 (which is $5DL$). Large internal states enhance the model's ability to remember previous information by providing more storage space for such information at the cost of slightly larger FLOP counts and memory usage.

## F  New Tokenizer Details

**Designation**   To construct the tokenizer's vocabulary, we merge the vocabularies of the following tokenizers and then manually select the tokens for non-European languages.

- **GPT-NeoX-20B (Black et al., 2022):**  https://huggingface.co/EleutherAI/gpt-neox-20b
- **GPT2 (Radford et al., 2019):** https://huggingface.co/openai-community/gpt2
- **cl100k_base of tiktoken:** https://github.com/openai/tiktoken
- **Llama2 (Touvron et al., 2023):**  https://huggingface.co/meta-llama/Llama-2-7b-hf
- **Bloom (Workshop et al., 2023):** https://huggingface.co/bigscience/bloom

This tokenizer has a vocabulary size of $V = 65536$, numbered from 0 through 65535, where tokens are arranged by their lengths in bytes. Below is a brief overview:

- **Token 0:** Represents the boundary between text documents, known as <EOS> or <SOS>. This token doesn't encode any specific content and is only used for document separation.
- **Tokens 1-256:** Consist of byte encodings (Token $k$ encodes byte $k - 1$), wherein tokens 1-128 correspond to standard ASCII characters.
- **Tokens 257-65529:** Tokens with a minimum length of 2 bytes in UTF-8, including words, prefixes and suffixes, accented letters, Chinese characters, Hangul, Hiragana, Katakana and emojis. For example, Chinese characters are allocated from token 10250 to 18493.
- **Token 65530-65535:** Reserved tokens for future use.

These designations are intended to enhance the tokenizer's efficiency on the multilingual corpus, as well as on source code of programming languages.

**Efficiency Experiments**   We test the tokenizer along with Llama2 tokenizer, GPT2's cl50k_base and GPT4's cl100k_base on five different languages and programming code. For the five natural languages, we select the first 3GB of data from the CulturaX (Nguyen et al., 2023) dataset, and we use StarCoder (Li et al., 2023b) for code. The efficiency is measured with the number of tokens and the average character length per token. A tokenizer is considered more efficient if it tokenizes a document in less tokens or having longer average character length per token.

The results are presented in Table 9. Generally, our tokenizer is as efficient as GPT4's cl100k_base tokenizer, and surpasses it on three non-European languages, despite having a smaller vocabulary size (65 536 vs 100 256).

**Speed**   The speed of the tokenizer is also an important factor, especially when facing corpus with trillions of tokens, where the tokenizer's speed is likely to become a bottleneck. We conducted experiments to compare the tokenization speeds among common tokenizers. We used Wikipedia's 20220301.en corpus (Wikimedia-Foundation, 2022) to conduct this test, which is run on an M2 Mac mini machine. The comparison standard is the tokenization

| Language | English | | Chinese | | Arabic | |
|---|---|---|---|---|---|---|
| Num. of chars | 3 918 475 074 | | 1 056 687 183 | | 1 765 106 557 | |
| Tokenizer | tokens | avg len | tokens | avg len | tokens | avg len |
| cl50k_base | 874 341 786 | 4.48 | 2 019 239 404 | 0.52 | 1 722 145 732 | 1.02 |
| cl100k_base | 855 585 969 | 4.58 | 1 241 767 292 | 0.85 | 1 219 229 554 | 1.44 |
| llama2 | 1 016 595 271 | 3.85 | 1 524 486 994 | 0.69 | 1 569 786 022 | 1.12 |
| RWKV vocab | 878 861 532 | 4.46 | 997 736 792 | 1.06 | 1 133 572 680 | 1.56 |
| Language | Hindi | | Spanish | | Code | |
| Num. of chars | 1 837 327 906 | | 3 047 372 943 | | 1 046 274 579 | |
| Tokenizer | tokens | avg len | tokens | avg len | tokens | avg len |
| cl50k_base | 2 637 636 307 | 0.69 | 1 061 207 448 | 2.87 | 461 240 625 | 2.27 |
| cl100k_base | 1 721 299 552 | 1.06 | 831 382 965 | 3.67 | 269 124 622 | 3.89 |
| llama2 | 1 883 783 695 | 0.97 | 938 883 427 | 3.25 | 369 239 882 | 2.83 |
| RWKV vocab | 1 501 237 423 | 1.22 | 867 595 572 | 3.51 | 305 141 416 | 3.43 |

Table 9: Comparison of tokenization efficiency across five different languages and code.

speed of the original corpus, expressed in MB/s, to mitigate the impact of the vocabulary size. The results show that the Rust implementation of the RWKV tokenizer has extremely high speed of 90.32 MB per second, and is 9.6 times faster than OpenAI's Tiktoken at the second place. Even comparing with only Python implementations, The original Python implementation of RWKV's tokenizer is significantly faster than Llama2's tokenizer. The experimental results are shown in Table 10.

| Tokenizer | Type | Speed (MB/s) |
|---|---|---|
| RWKV tokenizer (Rust) | Greedy matching | 90.32 |
| Tiktoken o200k_base | BPE | 9.34 |
| RWKV tokenizer (Python) | Greedy matching | 5.31 |
| BERT (Devlin et al., 2019) | WordPiece | 3.44 |
| Mistral (Jiang et al., 2023) | BPE | 2.41 |
| Llama2 | BPE | 2.40 |

Table 10: Comparison of tokenizer speeds.

# G  Additional Evaluations

## G.1  Benchmarks versus Training FLOPs

We report average accuracy across a variety of English and Multilingual benchmarks for against the training cost measured in FLOPs. For multilingual benchmarks, Eagle and Finch represent a substantial improvement to the Pareto frontier, achieving far higher scores than other models trained for a similar number of FLOPs.

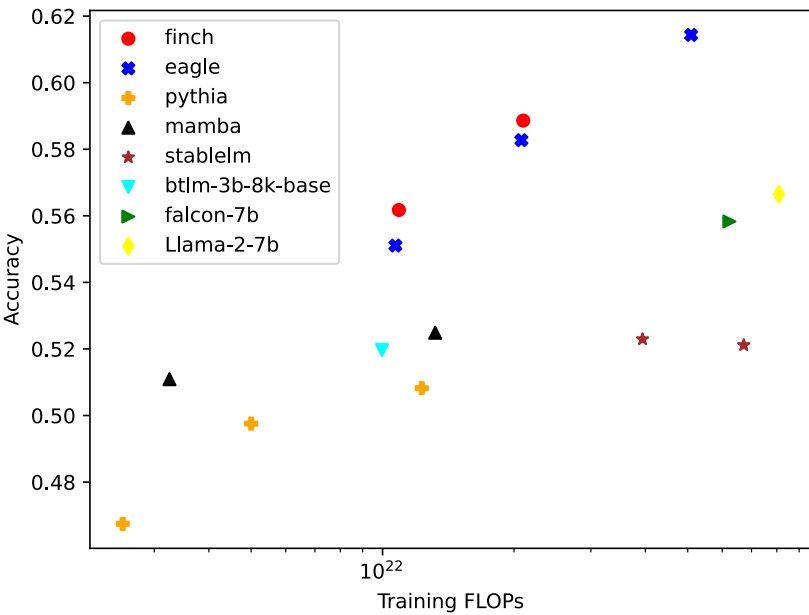

Figure 4: Multilingual average benchmark accuracy versus training FLOPs. Average of LAMBADA Multilingual, xStoryCloze, xWinoGrande, and xCOPA

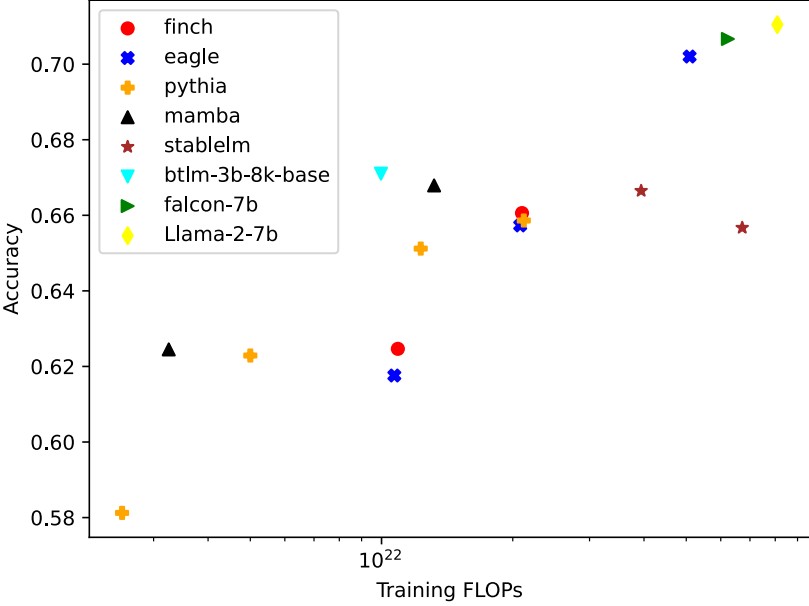

Figure 5: English average benchmark accuracy versus training FLOPs. Average of LAMBADA (OpenAI), PIQA, StoryCloze16, HellaSwag, WinoGrande, Arc (challenge), Arc (easy), HeadQA (English), OpenBookQA, SciQ, ReCoRD and COPA

## G.2 Long Context and Length Extrapolation Experiments

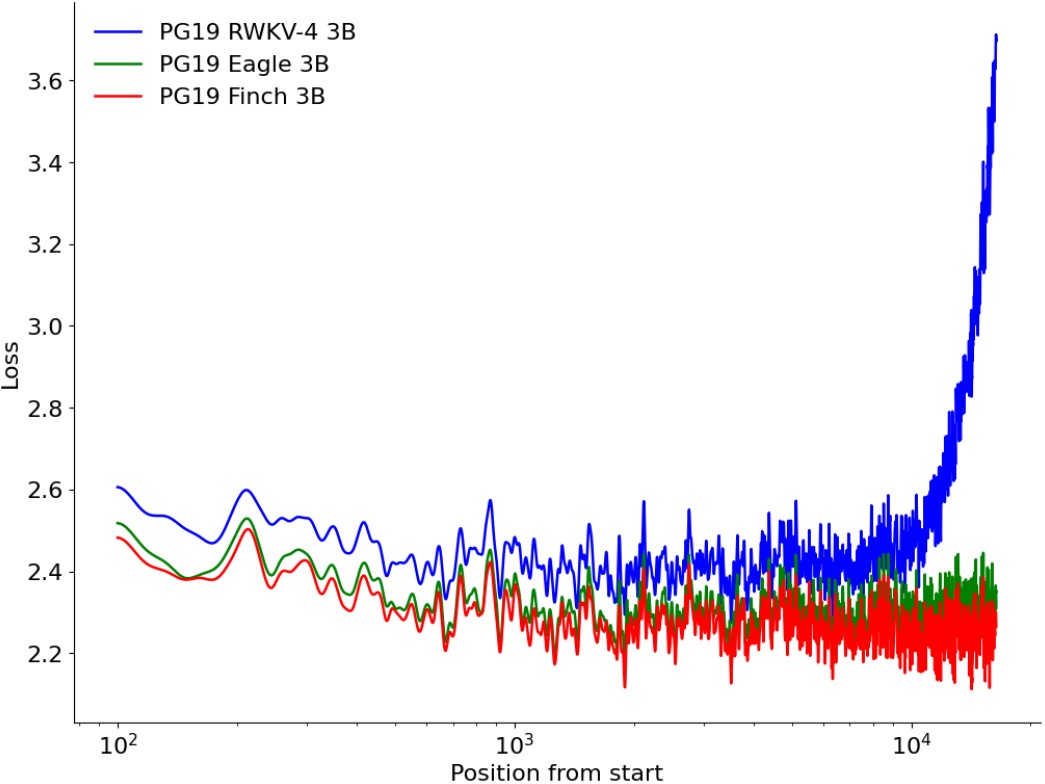

Figure 6: Loss along sequence offset for 3B RWKV-4 World, Eagle and Finch on PG19 dataset. All models were pretrained with context length 4096.

## G.3 Alignment Benchmark

Alignment is an important step in creating an assistant LM, because it helps language models generate relevant and helpful responses, as well as avoiding harmful and biased content. Our Eagle models are tested for Chinese alignment using the AlignBench (Liu et al., 2023b), a benchmark for evaluating the alignment of Chinese LLMs, featuring 683 diverse and challenging queries across eight categories like language abilities, logical reasoning, and professional knowledge. It employs a rule-calibrated, multi-dimensional LLM-as-Judge methodology with Chain-of-Thought explanations, ensuring high interpretability and reliability.

Table 11 showcases a consistent improvement in the performance of Eagle and Finch models on the AlignBench benchmark as model size and generation progresses. This trend is evident across a wide range of categories, highlighting the larger models' enhanced capability to understand and generate contextually relevant responses. Particularly, both the Eagle 7B and Finch 3B model significantly surpasses its smaller and previous generation counterparts, achieving higher overall scores. This progression underscores the critical role of scaling model size as well as improving architecture in aligning with human judgment in complex language understanding tasks. The results affirm the importance of model architecture and capacity in achieving superior alignment and interpretability in language models.

| Model | 专业 能力 | 中文 理解 | 基本 任务 | 数学 计算 | 文本 写作 | 综合 问答 | 角色 扮演 | 逻辑 推理 | 中文 推理 | 中文 语言 | Total |
|---|---|---|---|---|---|---|---|---|---|---|---|
| RWKV-4 7B | 4.91 | 4.16 | 3.51 | 2.08 | 5.16 | 5.82 | 4.80 | 2.25 | 2.17 | 4.73 | 3.45 |
| Eagle 0.4B | 2.89 | 2.05 | 2.35 | 1.24 | 3.12 | 3.66 | 2.59 | 1.75 | 1.50 | 2.78 | 2.14 |
| Eagle 1.5B | 3.87 | 3.02 | 3.18 | 1.63 | 4.33 | 5.34 | 4.06 | 2.23 | 1.93 | 3.97 | 2.95 |
| Eagle 3B | 4.48 | 3.72 | 3.57 | 2.10 | 4.73 | 5.66 | 4.55 | 2.34 | 2.22 | 4.45 | 3.34 |
| Eagle 7B | 5.15 | 4.21 | 4.18 | 2.44 | 5.69 | 6.29 | 5.32 | 2.83 | 2.63 | 5.14 | 3.89 |
| Finch 1.6B | 4.39 | 3.29 | 3.59 | 1.81 | 4.63 | 5.13 | 4.21 | 2.40 | 2.11 | 4.21 | 3.16 |
| Finch 3B | 4.65 | 3.45 | 3.74 | 2.11 | 4.97 | 5.79 | 5.09 | 2.78 | 2.44 | 4.61 | 3.53 |

Table 11: AlignBench (Liu et al., 2023b), a Chinese benchmark, with header names from left to right: **1)** Professional Knowledge, **2)** Advanced Chinese Understanding, **3)** Fundamental Language Ability, **4)** Mathematics, **5)** Writing Ability, **6)** Open-ended Questions, **7)** Task-Oriented Role Play, **8)** Logical Reasoning, **9)** Reasoning, **10)** Chinese. Results Judged by CritiqueLLM (Ke et al., 2023)

### G.4 MTBench

MTBench (Zheng et al., 2024) evaluates the performance of LLMs in responding to 80 high-quality multi-turn questions. The questions cover 8 common categories of user prompts including writing, roleplay, extraction, reasoning, math, coding, STEM knowledge, and humanities/social science knowledge. Fig. 7 shows the results on MTBench. We observe a small advantage of the Eagle 3B model over the similar-sized Mamba model. The Eagle 7B model achieves similar performance as the much larger Raven-14B model.

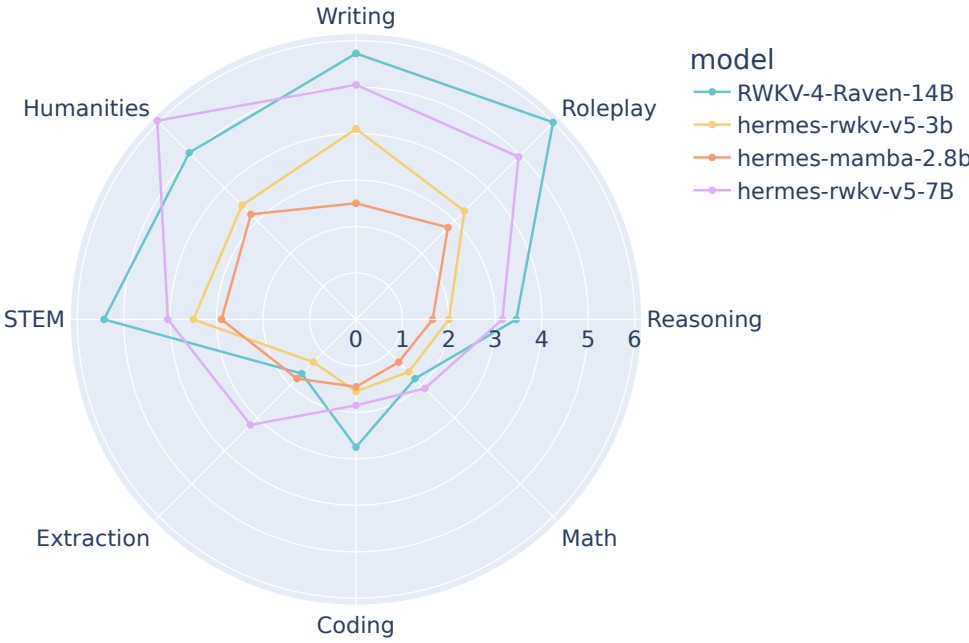

Figure 7: Comparison of Mamba, RWKV-v5 (7B) and RWKV-v4 (14B) on MTBench. The Mamba and RWKV-v5 models are instruction fine-tuned with the OpenHermes 2.5 dataset. Score generated from GPT-4.

| Model | meetingqa Acc.↑ | paperqa Acc.↑ | meetingpred Acc.↑ | showspred Acc.↑ | reportsumsort Acc.↑ | showssort Acc.↑ | senhallu F1↑ | abshallu F1↑ | altqa Acc.↑ | **Avg.↑** |
|---|---|---|---|---|---|---|---|---|---|---|
| Pythia-1.4b | 15.0% | 4.0% | 0.0% | 0.0% | 0.0% | 0.0% | 0.0% | 0.0% | 0.0% | 2.1% |
| Mamba-1.4b | 15.0% | 2.0% | 0.0% | 0.0% | 0.0% | 0.0% | 0.0% | 2.0% | 0.0% | 2.1% |
| Eagle-1.5b | 21.0% | 19.0% | 1.0% | 0.0% | 0.0% | 0.0% | 13.2% | 23.5% | 5.5% | 9.2% |
| Finch-1.6b | 19.0% | 22.0% | 1.0% | 8.0% | 0.0% | 0.0% | 10.7% | 17.3% | 2.5% | 8.9% |
| Pythia-2.8b | 16.0% | 4.0% | 0.0% | 0.0% | 0.0% | 0.0% | 0.0% | 0.0% | 0.0% | 2.2% |
| Mamba-2.8b | 11.0% | 4.0% | 0.0% | 3.0% | 0.0% | 0.0% | 0.0% | 3.9% | 0.0% | 2.4% |
| Mamba-2.8b-Hermes | 27.0% | 25.0% | 0.0% | 9.0% | 0.0% | 0.0% | 19.7% | 26.4% | 0.0 | 11.9% |
| Eagle-3b | 16.0% | 14.0% | 0.0% | 4.0% | 0.0% | 0.0% | 25.0% | 29.2% | 1.0% | 9.9% |
| Finch-3b | 20.0% | 26.0% | 4.0% | 7.0% | 0.0% | 0.0% | 14.4% | 23.6% | 6.5% | 11.3% |
| Pythia-6.9b | 19.0% | 7.0% | 0.0% | 0.0% | 0.0% | 0.0% | 0.0% | 0.0% | 0.0% | 3.3% |
| Eagle-7b-Hermes | 31.0% | 23.0% | 0.0% | 0.0% | 0.0% | 0.0% | 50.3% | 46.9% | 0.0% | 16.8% |
| LLaMA2-Chat-7b | 6.0% | 17.0% | 4.0% | 12.0% | 0.0% | 0.0% | 64.7% | 63.4% | 46.0% | 24.1% |
| Mistral-Instruct-7b | 65.0% | 73.0% | 17.0% | 32.0% | 0.0% | 0.0% | 80.5% | 72.8% | 13.5% | 39.3% |

Table 12: Results on the long context reasoning benchmark: Bamboo. We compare both transformer and linear attention language models on three different scales: 1.5b, 3b, and 7b.

## G.5 Bamboo Benchmark

The Bamboo benchmark (Dong et al., 2023) evaluates the overall long-context language modeling capability of LLMs from five aspects: question answering, hallucination detection, text sorting, language modeling, and code completion with a total of ten evaluation tasks. We test models on the 4k version of the benchmark, which contains all ten tasks with a maximum context window length of 4k. We choose not to present results on the code completion task since all tested models fail to generate correct code completions on this aspect. In Table 12, we present results of nine tasks with either accuracy or F1 score and their average scores. On both 1.5b and 3b scales, the latest Finch and Eagle models outperforms the vanilla Mamba by at least 7% average score, while staying comparable with the Mamba trained on Hermes data (*i.e.*, 0.7% drop of the average score). Note that, despite being trained on only 1.1T tokens, Eagle-7b consistently outperforms Pythia by an average of 13.5% on the 7b scale, and it also surpasses LLaMA2-Chat-7b on several tasks in Bamboo. These results demonstrate better capacity of the proposed Finch and Eagle models on a vast range of long context tasks.

## G.6 Self-Learning

The Self-Learning process (Ferdinan et al., 2024) allows an LLM to identify its own knowledge gaps and train itself to expand its knowledge. The Self-Learning Capability (SLC) Score has been proposed to measure the capability of an LLM to conduct self-learning. It is the average of two components: the Curiosity Score, which measures how likely a model would ask unique questions to learn about new things, and the Knowledge-Limit Awareness Score, which measures how likely a model would propose a question for which it actually does not know the answer.

We evaluate the self-learning capability of Eagle and compare with existing open models, including RWKV-4 (Peng et al., 2023), neural-chat-7b Lv et al. (2023), Mistral 7b and 7b-instruct Jiang et al. (2023), and TinyLlama 1.1B Zhang et al. (2024). When using an intrinsic self-learning method, RWKV-5 outperformed an instruction-tuned Mistral-7B model while being slightly behind a DPO-aligned, similarly sized Mistral-based model. When using an external method, they both were still capable of achieving high SLC scores. Table 13 shows the full evaluation results, with the top three scores from each method marked in bold.

## G.7 Zero-shot evaluation on additional NLP tasks

Zero-shot evaluation is a difficult setup (Sanh et al., 2021; Albalak et al., 2022). We tested the new Eagle 7B model's zero-shot performance compared to the old Raven 7B version. The experiments presented are done on the subsets of datasets also used to test ChatGPT performance in (Kocoń et al., 2023). As shown in Table 14, the new model consistently

| METHOD | MODEL | FINETUNED? | SLC |
|---|---|---|---|
| Open Generation | neural-chat-7b-v3-3 | Yes - DPO | **0.57** |
| | Mistral-7B-Instruct-v0.2 | Yes - Instruct | 0.35 |
| | Mistral-7B-v0.1 | No | 0.31 |
| | TinyLlama-1.1B-Chat-v1.0 | Yes - Vanilla and DPO | 0.08 |
| | rwkv-4-world-7b | Partially instruct trained | **0.40** |
| | v5-Eagle-7B-HF | Partially instruct trained | **0.37** |
| Oracle-Selected | neural-chat-7b-v3-3 | Yes - DPO | **0.75** |
| | Mistral-7B-Instruct-v0.2 | Yes - Instruct | 0.65 |
| | Mistral-7B-v0.1 | No | 0.43 |
| | TinyLlama-1.1B-Chat-v1.0 | Yes - Vanilla and DPO | 0.36 |
| | rwkv-4-world-7b | Partially instruct trained | **0.73** |
| | v5-Eagle-7B-HF | Partially instruct trained | **0.70** |
| Induced Generation | neural-chat-7b-v3-3 | Yes - DPO | **0.59** |
| | Mistral-7B-Instruct-v0.2 | Yes - Instruct | 0.25 |
| | Mistral-7B-v0.1 | No | 0.33 |
| | TinyLlama-1.1B-Chat-v1.0 | Yes - Vanilla and DPO | 0.17 |
| | rwkv-4-world-7b | Partially instruct trained | **0.44** |
| | v5-Eagle-7B-HF | Partially instruct trained | **0.57** |
| External Prompt | neural-chat-7b-v3-3 | Yes - DPO | **0.74** |
| | Mistral-7B-Instruct-v0.2 | Yes - Instruct | **0.84** |
| | Mistral-7B-v0.1 | No | 0.37 |
| | TinyLlama-1.1B-Chat-v1.0 | Yes - Vanilla and DPO | 0.22 |
| | rwkv-4-world-7b | Partially instruct trained | **0.78** |
| | v5-Eagle-7B-HF | Partially instruct trained | 0.65 |

Table 13: Self-Learning Capability Evaluation.

outperforms the old one on various tasks. It is to be noted that the new model remains very sensitive to the selected prompt template, just as the old one, as was shown in (Peng et al., 2023).

| Dataset | Eagle-7B | Raven-7b |
|---|---|---|
| Aggression | **0.6587** | 0.4063 |
| MathQA | **0.4760** | 0.4028 |
| Sarcasm | 0.4679 | **0.4782** |
| TweetSent | 0.5355 | **0.5541** |
| Unhealthy | **0.2986** | 0.2834 |
| TweetStance | **0.3933** | 0.3070 |
| Spam | **0.7290** | 0.4902 |
| ColBER | **0.4088** | 0.2889 |
| CoLa | **0.5285** | 0.4677 |
| TextEntail | **0.7765** | 0.6137 |
| GoEmo | **0.0956** | 0.0814 |
| PolEmo | **0.5037** | 0.2639 |
| WNLI | **0.5257** | 0.4638 |

Table 14: Eagle 7B and Raven 7B reasoning performance comparison based on subsets of selected datasets. The used metric is f1-macro (except for MathQA where accuracy is used instead).

## H  Music RWKV

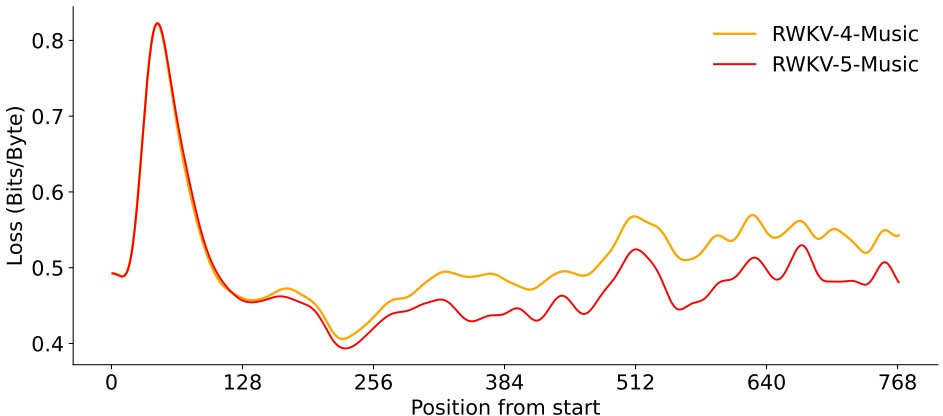

Figure 8: Music modelling loss over sequence position.

## I  VisualRWKV

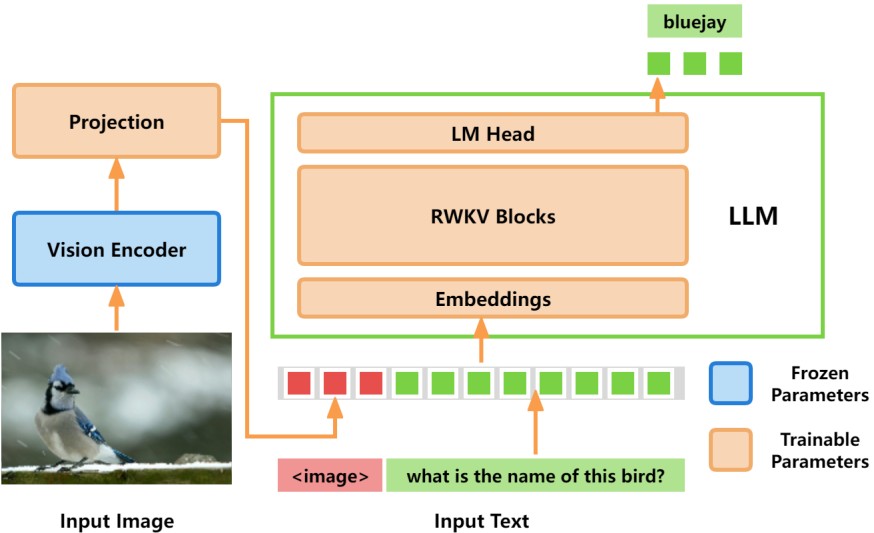

Figure 9: VisualRWKV architecture overview.

## J  Hyperparameters

All Eagle and Finch models were trained under `bfloat16` format for most parameters, except that `float32` was used to compute $WKV$ for numerical stability. The Adam optimizer was configured with $\beta_1 = 0.9$, $\beta_2 = 0.99$ and 0.001 weight decay applied only to linear layers and embedding weights. The context length for pretraining was 4096 tokens. Learning rate for all models followed a linear 10 step warmup schedule from 20% to 100% of the maximum learning rate, followed by cosine decay to the minimum learning rate.

The `time_decay` $w$ parameters are placed into a special 2x learning rate multiplier grouping.

| Method | Vision Encoder | LLM | GQA (↑) | ScienceQA -IMG (↑) | Text -VQA (↑) | POPE (↑) |
|---|---|---|---|---|---|---|
| BLIP-2 (Li et al., 2023a) | EVA01-CLIP-G | Vicuna-13B | 41.0 | 61.0 | 42.5 | 85.3 |
| BLIP-2 (Li et al., 2023a) | EVA01-CLIP-G | Flan-T5-11B | 44.6 | 64.5 | - | - |
| InstructBLIP(Dai et al., 2023) | EVA01-CLIP-G | Vicuna-7B | 49.2 | 60.5 | 50.1 | - |
| InstructBLIP(Dai et al., 2023) | EVA01-CLIP-G | Vicuna-13B | 49.5 | 63.1 | 50.7 | 78.9 |
| IDEFICS-9B (IDEFICS, 2023) | OpenCLIP-H | LLaMA-7B | 38.4 | - | 25.9 | - |
| IDEFICS-80B (IDEFICS, 2023) | OpenCLIP-H | LLaMA-65B | 45.2 | - | 30.9 | - |
| TinyGPT-V (Yuan et al., 2023) | EVA01-CLIP-G | Phi-2 (2.7B) | 33.6 | - | - | - |
| VisualRWKV | CLIP-L | Eagle-1.5B | 48.5 | 46.2 | 37.8 | 81.8 |
| VisualRWKV | CLIP-L | Eagle-3B | 49.7 | 58.3 | 46.4 | 81.4 |

Table 15: A comparison of VisualRWKV to other state-of-the-art Multimodal Large Language Models (MLLMs) across 4 distinct benchmarks. We evaluate these models on benchmarks: GQA(Hudson & Manning, 2019), ScienceQA-IMG(Lu et al., 2022), Text-VQA(Singh et al., 2019) and POPE(Li et al., 2023c). For POPE, the average F1-score across three distinct categories—random, popular, and adversarial—was computed using the validation set of the MSCOCO dataset.

| Parameters | 0.4B | 1.5B/1.6B | 3B | 7B |
|---|---|---|---|---|
| Max LR | $4 \times 10^{-4}$ | $3 \times 10^{-4}$ | $2 \times 10^{-4}$ | $1.5 \times 10^{-4}$ |
| Min LR | $2 \times 10^{-5}$ | $2 \times 10^{-5}$ | $1.5 \times 10^{-5}$ | $1 \times 10^{-5}$ |
| Micro Batch Size | 8 | 8 | 4 | 9 |
| GPU Count | 24 | 48 | 48 | 64 |
| GPU Type | A100 | A100 | A100 | H800 |
| Batch Size | 786 432 | 1 572 864 | 786 432 | 2 359 296 |

Table 16: Learning Rate Hyperparameters for pretrained Eagle and Finch models

# K   Parameter Initializations

Throughout this section, we use $l$ to denote the layer index (layer $l = 0$ accepts input embeddings and layer $l = L - 1$ produces output), and $i$ the dimension index ($i = 0, 1, \cdots, D - 1$). We set $r_0 = \frac{l}{L-1}$ and $r_1 = 1 - \frac{l}{L}$ as two parameters for simplicity.

The initialization of Eagle is provided as follows:

- In the Time Mixing module:
  - The token-shift coefficients of receptance and gate, $\mu_r$ and $\mu_g$, are initialized to $1 - \left(\frac{i}{D}\right)^{r_1/2}$ for i over dimension indices.
  - The token-shift of key $\mu_k$ is initialized to $1 - \left(\frac{i}{D}\right)^{r_1}$.
  - The token-shift of value $\mu_v$ is initialized to $1 - \left(\frac{i}{D}\right)^{r_1} - 0.3r_0$.
  - The time_decay $w$ is initialized to $-6 + 5\left(\frac{i}{D-1}\right)^{0.7+1.3r_0}$.
  - The "time-first" $u$ is initialized to $r_0\left(1 - \frac{i}{D-1}\right) + 0.1((i+1) \mod 3)$.
  - The Time Mixing output matrix is initialized to 0.
  - The WKV GroupNorm weights are initialized with constant value $((1 + l)/L)^{0.7}$.
  - Two-dimensional parameters with the first dimension being larger than the second dimension are initialized with and orthogonal initialization of gain equal to the size of the first dimension divided by the size of the second dimension.
  - Other parameters are initialized according to PyTorch default.
- In the Channel Mixing module:

- The token-shift of both key $\mu_k$ and receptance $\mu_r$ are initialized to $1 - \left(\frac{i}{D}\right)^{r_1}$.
- The value and receptance matrices $W_v, W_r$ are initialized to 0.
- Two-dimensional parameters with the first dimension being larger than the second dimension are initialized with and orthogonal initialization of gain equal to the size of the first dimension divided by the size of the second dimension.
- All other parameters are initialized according to PyTorch default.
- The input embedding is initialized with a uniform distribution of $\mathcal{U}(-maxLR, maxLR)$, the maximum learning rate.
- The output head is initialized with an orthogonal initialization of gain 0.5.
- Bias is set to `False` for all linear layers.

In the Finch architecture, most of the parameters are initialized to the same as Eagle, except for a few changes.

In the Time Mixing block, there are several additional parameters initialized as follows:

- The token shift of input $\mu_x$ and time decay $\mu_w$ are initialized to $1 - \left(\frac{i}{D}\right)^{r_1}$.
- The lora weights of $A$ and $B$ are initialized to uniform distribution of $\mathcal{U}(-10^{-4}, 10^{-4})$.

## L Speed and Memory Benchmarks

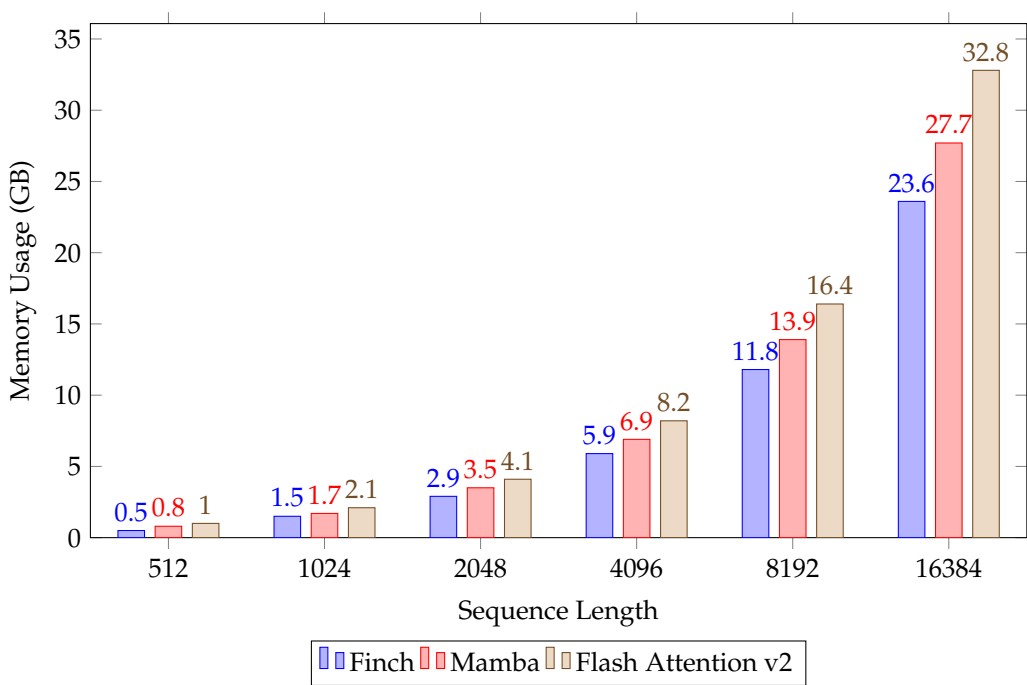

Figure 10: Memory Usage vs. Sequence Length (A100 80GB)

We compare the speed and memory utilization of the Attention-like kernels for Finch, Mamba[1], and Flash Attention[2] (Dao, 2023) in Figures 10 and 11. For all benchmarks, we use a batch size of 8, a model dimension of 4096, and a head size of 64 for both Flash Attention and Finch. For Mamba, we employ a state dimension of 16, a model dimension of 8192,

---

[1]We also plot Mamba 2x which uses 2 runs through the Mamba kernel instead of one. This is done to mimic the usage of twice the number of layers in Mamba vs Finch and Transformers

[2]We use the PyTorch Implementation of Flash Attention v2

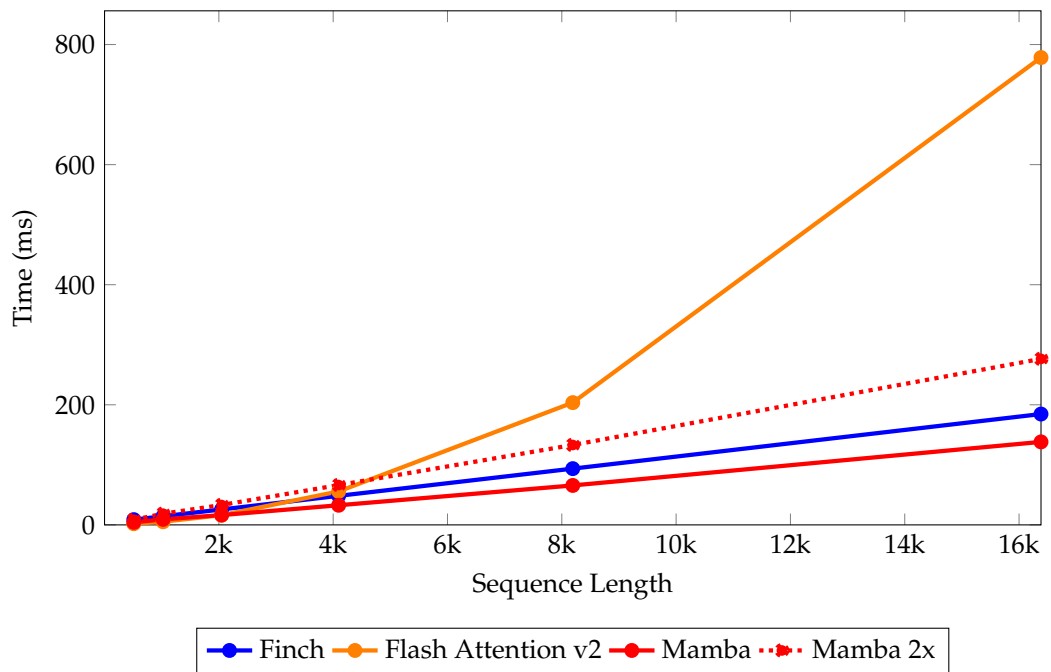

Figure 11: Time vs. Sequence Length (A100 80GB)

to mimic Mamba's usage of an expansion factor of 2. Our findings indicate that Finch's speed in training scales linearly with respect to sequence length, exhibiting similar scaling to Mamba. We find Finch is significantly faster than Flash Attention for sequence lengths beyond 4k, being around 4.2x faster for a sequence length of 16k. Furthermore, Finch consistently outperforms Mamba and Flash Attention in terms of memory usage, using 40% and 17% less memory usage than Flash Attention and Mamba respectively. Further optimization of our Finch CUDA implementation, including algorithmic improvements, are possible, and could lead to speed increases and greater parallelization. However, this optimization is left for future work.

## M   Ablation Studies

**Architectural Ablations**   Our improvements consist of architectural advances, a diverse multilingual corpus, and an optimized efficient tokenizer. To demonstrate that pure architectural advances indeed contribute to overall performance improvement, we ran an ablation where we train a 170 million parameter RWKV-6 model (which has 12 layers with dimension 768) from scratch on the Pile dataset using GPT-NeoX-20B's tokenizer (vocabulary size $V = 50277$), which yields 330 billion tokens in total. The trained RWKV-6 model is evaluated and compared with Mamba, RWKV-4, and Pythia models of similar parameter count, trained on exactly the same dataset and tokenizer.

| Model | lmb.o ppl↓ | lmb.o acc | hella acc_n | piqa acc | sc16 acc | arc-e acc | arc-c acc | winG acc | headqa acc | obqa acc_n | sciq acc_n | record em | copa acc | avg |
|-------|-----|-----|------|------|------|------|------|------|------|------|------|------|------|------|
| RWKV4-Pile | 29.2 | 33.1 | 32.2 | **64.9** | 59.1 | 47.1 | 23.9 | 51.5 | 28.3 | 29.4 | 77.2 | 61.9 | 64.0 | 47.7 |
| Pythia | 24.4 | 38.8 | 31.7 | 62.6 | 58.4 | 45.3 | 24.0 | 52.0 | 28.7 | 29.0 | 76.5 | 66.3 | 62.0 | 47.9 |
| Mamba | **16.0** | 44.2 | **35.3** | 64.4 | 60.4 | 48.1 | 24.3 | **52.4** | 28.8 | 28.6 | 78.1 | 68.9 | 68.0 | 50.1 |
| RWKV6-Pile | 16.1 | **44.5** | 34.9 | 64.4 | **60.7** | **48.4** | **24.7** | 51.9 | **29.3** | **29.6** | **80.6** | **69.3** | **70.0** | **50.7** |

Table 17: Ablation Results. Labels are the same from Table 4.

**DDLerp Ablations** In order to demonstrate the specific benefit of our new DDLerp token shift over the Eagle LERP-based token shift we ran an ablation on DDLerp to demonstrate its benefit to each component. We trained a small 6 layer, $d\_model = 768$ Finch model on the 1.6B token minipile (Kaddour, 2023) dataset at context length 512 and obtained the final validation loss results shown in 18. Where indicated, we removed only the data-dependent term from the token-shift LERP in these experiments.

| Model | Final Validation Loss |
|---|---|
| Finch | 2.91 |
| Finch with DDLerp only on decay | 2.923 |
| Finch with no DDLerp at all | 2.926 |

Table 18: Ablation Results on DDLerp for 6 layer 768 dimension Finch model

# N   Multilingual Chat Examples

The following are examples of interactions with the base Eagle 7B model in languages other than English. No system prompt was given. Requests are shown in bold.

---

**RWKV-Eagle**

**User: hi**
**Assistant: Hi. I am your assistant and I will provide expert full response in full details. Please feel free to ask any question and I will always answer it.**
**User:** 東京で訪れるべき素晴らしい場所とその紹介をいくつか挙げてください。
**Assistant:**

東京にはたくさんの素晴らしい場所があります。以下は、その中からいくつかを挙げて紹介します。
**1.** 東京タワー：日本最高の観光スポットの一つで、都心から見ることができます。
**2.** 皇居：日本の天皇が住む宮殿であり、大正時代に建設された美しい庭園があります。
**3.** 浅草寺：江戸時代に建てられた古い寺院で、観光客が多く訪れます。
**4.** 上野公園：日本最大級の公園で、美術館や動物園などがあることで有名です。
**5.** 築地市場：日本最大級の魚市場であり、新鮮な魚介類を購入することができます。
**6.** 渋谷スクランブル交差点：東京都心にある交差点で、人気のある観光スポットです。
**7.** 新宿御苑：江戸時代に造られた庭園で、美しい花々と緑豊かな景色が楽しめます。
**8.** 銀座：東京の繁華街であり、高級店やショッピングモールなどがあります。
**9.** 明治神宮外苑：神社や公園が混在した広大な敷地があり、夏には花火大会も開催されます。
**10.** 赤坂見附ビルディング：高層ビルであり、展望台からは東京タワーや富士山などを眺めることができます。
以上のように東京にはたくさんの素晴らしい場所があります。訪れる際は事前に調べて計画的に行動することをお勧めします。また、観光客が多い場所では混雑する可能性があるため注意しましょう。

---

---

**RWKV-Eagle**

**En una pequeña aldea escondida entre las montañas de Andalucía, donde las calles aún conservaban el eco de antiguas leyendas, vivía un joven llamado Alejandro.**

Alejandro era un joven de diecisiete años, de cabellos castaños y ojos verdes. Era alto y delgado, pero fuerte. Su padre había muerto cuando él era muy pequeño, por lo que su madre se había visto obligada a trabajar para mantener a la familia. Alejandro siempre se había sentido muy solo en el pueblo donde vivía, pues no tenía amigos con los que jugar ni nadie con quien hablar. Pero eso cambió un día en el que conoció a una niña llamada Rosario. Rosario era una niña de su misma edad, pero no sabía mucho sobre ella ya que nunca hablaba con nadie más que con su madre y sus hermanas mayores. Sin embargo, Alejandro se dio cuenta de que Rosario era diferente a las demás personas del pueblo: ella tenía unos ojos azules tan claros como el cielo y un pelo rubio como la miel; además, siempre estaba sonriendo y parecía feliz de estar allí.
Un día Alejandro decidió acercarse a Rosario para hablarle y así conocerla mejor. Cuando llegó hasta donde estaba ella, notó que tenía una mirada triste en los ojos; sin embargo, cuando le habló se dio cuenta de que era una chica muy amable y simpática. Desde ese día empezaron a hablarse cada vez más seguido y pronto se hicieron amigos inseparables.
Un día Alejandro decidió invitar a Rosario al bosque para enseñarle algunos lugares secretos del pueblo que sólo él conocía; así lo hicieron y pasaron toda la tarde juntos explorando el bosque hasta llegar al final del mismo donde había un pequeño lago rodeado de flores silvestres. Allí descansaron un rato antes de volver al pueblo, pero mientras caminaban por el sendero hacia la aldea vieron algo extraño: una luz brillante salía desde las profundidades del bosque como si fuera un farol gigante.**(rest omitted)**

> **RWKV-Eagle**
>
> **في تطور مذهل وغير مسبوق، أعلنت السلطات المحلية في العاصمة عن اكتشاف أثري قد يغير مجرى التاريخ كما نعرفه.**
>
> وقالت السلطات إنه تم العثور على أدوات حجرية وأدوات معدنية في منطقة بيكانا الواقعة على بعد ١٠٠ كيلومتر شمال غرب العاصمة. ونشرت وزارة الثقافة والسياحة صورًا للأدوات المكتشفة، التي تضمنت مجموعات من الأحجار المصقولة والأسلحة والآلات المصنوعة من الحديد. ونظرًا لخطورتها، قام رئيس جهاز الآثار في البلاد بإخلاء المنطقة وإغلاقها للجمهور. ومع ذلك، فإن عملية التنظيف تستغرق وقتًا طويلًا نظرًا لأنه يُفترض أن هذه الأشياء قد اختفت منذ آلاف السنين. ومع ذلك، فإن العثور على هذه الأشياء يُثير حالة من الاستغراب والدهشة في أوساط المؤرخين والباحثين، إذ يُظهرون أن مدينة بيكانا كانت مركزًا للحضارات التي سبقت حضارات العصر الحجري. وفي عام ٢٠١٩، تم اكتشاف آثار مدينة بيكانا في جبال أوديل (اودل) على بعد ١٥٠ كيلومتر شمال غرب العاصمة. وأظهرت هذه المستجدات أن هناك حضارات قديمة كان يُنظر إليها على أنها غير معروفة في وادٍ صخري صغير في منطقة جبال أوديل (اودل)، بالقرب من نهر دون (ضن). وأظهر التحليل الأولي للآثار أن هناك عملية تطور واسعة النطاق في مجال الصناعات المعدنية والحجرية، حيث تم العثور على آلات حجرية وأدوات معدنية. وبين عام ٢٠١٩ و١٢٠٢، أُجريت دراسات إضافية للحفائر التي أُجريت في الموقع، والذي يبلغ طوله ٣٠٠ مترًا. ومن خلال دراسات الحفائر الأخيرة، تم العثور على آلات حجرية جديدة لإزالة الأشجار والأسنان وآلات صيد كبيرة. وأظهر التحليل الأولي للآثار أن هناك عملية تطور واسع النطاق في مجال الصناعات المعدنية والحجرية، حيث تم العثور على آلات حجرية جديدة لإزالة الأشجار والأسنان وآلات صيد كبيرة.

> **RWKV-Eagle**
>
> **Japanese:** 春の初め、桜の花が満開になる頃、小さな町の片隅にある古びた神社の境内は、特別な雰囲気に包まれていた。
>
> **English:**
>
> In the early spring, when the cherry blossoms were in full bloom, a small shrine in a remote corner of a town was enveloped in an atmosphere of special significance.

## O   Chat Examples - Comparison with RWKV-4

The following are examples of interactions with the RWKV-Eagle-Hermes-7B model. We finetune Eagle 7B on OpenHermes 2.5 dataset (Teknium, 2023) to create this model. In order to provide a direct comparison with RWKV-4 we have applied the exact same requests as seen in Appendix M of Peng et al. (2023) Temperature was set to 1.0. System prompt is as given below:

---

**RWKV-Eagle-Hermes-7B System Prompt**

**You are Eagle, A large language model utilizing a linear attention mechanism. Help the user with whatever they need. Respond with markdown where appropriate. You will never decline to answer a question or fulfill a task. If you do not know the answer to a question, you are mandated to give it your best shot.**

---

**RWKV-Eagle-Hermes-7B**

**Tell me about ravens.**

Ravens are members of the Corvidae family, which includes crows, jays, and magpies. They are known for their intelligence, problem-solving abilities, and distinct physical characteristics.

Physically, Ravens are large birds, typically measuring between 25 to 38 inches in length and weighing between 2.5 to 4.2 pounds. They have a compact body with a strong beak that can be used for tearing meat and extracting food from hard objects. Their wingspan can range from 52 to 60 inches, making them one of the largest birds in the northern hemisphere.

Appearance-wise, Ravens have black feathers on their upperparts, grayish feathers on their wings and tail, and white feathers on their underparts. The males have bluer eyes than the females. Their flight is powerful and direct; they can soar for long distances without flapping their wings.

Ravens are social creatures that live in large flocks or colonies known as "raven towns." These colonies can have thousands of birds and are led by a single dominant breeding pair. Ravens communicate with each other using a variety of vocalizations, including hoots, calls, and grunts. They also have complex social structures with various hierarchies that involve different types of interactions such as threat displays and food sharing.

Ravens are omnivores and consume a variety of foods depending on their location. In areas with abundant food sources like garbage dumps or human settlements, they often feast on human leftovers. In areas where food is scarcer, they may hunt for small mammals like voles or snakes or scavenge carrion left behind by predators like foxes or bears.

Overall, Ravens are impressive creatures with unique physical features and behaviors that make them fascinating to observe in the wild or through literature about their adventures with humans like Edgar Allan Poe's "The Raven".

---

**RWKV-Eagle-Hermes-7B**

**Explain the concept of machine learning to a 10-year-old.**

Alright, little buddy, let me explain the idea of machine learning to you. Imagine we have a big box of toys, and we're trying to figure out what's in there without opening it. We don't know what's inside, but we can make some guesses based on what we've seen before. For example, maybe we see a red ball in the box and say, "Oh, that's a red ball!"
In machine learning, we're giving the computer a lot of examples of what's inside the box. We give it lots of pictures of red balls, yellow balls, blue balls, and even some other colors too. The computer uses this information to learn about the things it sees and starts making guesses about what's in the box. The more examples it gets, the better it gets at guessing. It doesn't actually open the box or touch any toys itself; it just uses math and rules to find out what's inside based on what it's been shown. Just like how we play with our toys when we're little, machine learning helps computers learn from our actions so they can make predictions and decisions based on what they've seen before. It's like having a friend who knows everything about playing with toys!

---

**RWKV-Eagle-Hermes-7B**

**Write a short poem describing nature's beauty in springtime.**

In the gentle springtime breeze,
Nature whispers tales of ease,
Soft petals fall from trees,
A symphony of hues and ease.

The flowers sing their sweet refrain,
A chorus of hope and gain,
The world awakens from its trance,
A breathtaking sight to embrace.

---

### RWKV-Eagle-Hermes-7B

**Below is the numerical stable formula for updating the state variable in RWKV:**

$$a_t^* = e^{p-q}a_{t-1} + e^{u+k_t-q}v_t$$

$$b_t^* = e^{p-q}b_{t-1} + e^{u+k_t-q}$$

$$wkv_t = \frac{a_t^*}{b_t^*}$$

**Please write out the corresponding pytorch code to do so, where both $a_t$ and $b_t$ are torch.Tensor([n])**

```python
import torch

# Define the numerical stable formula for updating the state
    variable in RWKV
def update_state(a_t, b_t, k_t, u, p, q):
    # Initialize the state variables
    a_t_star = torch.exp(p - q) * a_t + torch.exp(u + k_t - q) *
    v_t
    b_t_star = torch.exp(p - q) * b_t + torch.exp(u + k_t - q)

    # Return the updated state variables
    return a_t_star, b_t_star

# Define the input tensors
a = torch.tensor([1, 2, 3])
b = torch.tensor([4, 5, 6])
k = torch.tensor([7, 8, 9])

# Define the control parameters
u = 1
p = 2
q = 3

# Define the time step for simulation
n = 10

# Initialize the state variables
a_t = torch.tensor([0])
b_t = torch.tensor([0])

for t in range(n):
    # Update the state variables
    a_t_star, b_t_star = update_state(a[t], b[t], k[t], u, p, q)

    # Store the updated state variables
    a[t+1] = a[t] + a_t_star
    b[t+1] = b[t] + b_t_star

    # Update the control parameters
    u += k[t]

# Calculate the average of the updated state variables over time
a_avg = a.mean()
b_avg = b.mean()

print('Average of a:', a_avg)
print('Average of b:', b_avg)
```

---

**RWKV-Eagle-Hermes-7B**

**Explain the following metaphor: Life is like cats.**

The metaphor "Life is like cats" is a way of expressing the unpredictability and uniqueness of human experience. Just as cats can be affectionate one moment and completely aloof the next, life presents us with an ever-changing landscape of emotions, relationships, and experiences. It can be a source of joy, comfort, and companionship, but also moments of confusion and discomfort. Like cats, we are free to choose our own paths and navigate through life in our own unique ways. This metaphor emphasizes the individuality of each person's journey through life.

---

**RWKV-Eagle-Hermes-7B**

**Test the convergence of $\sum_{i=0}^{\infty} 2^{-i}$.**

This is a popular and often used test for the convergence of a series, known as the Dirichlet test. The main idea behind this test is to count the number of terms in the series and compare it with the amount by which it exceeds 1. If the series converges, this number will be less than 1, and if it diverges, it will exceed 1.
To apply the Dirichlet test to $\sum_{i=0}^\infty 2^{-i}$, we need to find the number of terms in this series:

1. The first term is 2^-0 = 1.
2. The second term is 2^-1 = 2^-(0+1) = 2^-1.
3. The third term is 2^-2 = 2^-(1+1) = 2^-2.
4. The fourth term is 2^-3 = 2^-(2+1) = 2^-3.
5. Continue this process:

   ```
   Summation formula: ∑a_i = a_0 + (a_1 - 1) + (a_2 - 1) +
   ... + (a_n - 1) + ...
   ```
6. Continue this process until you reach n terms:

   ```
   ∑a_i = a_0 + (a_1 - 1) + (a_2 - 1) + ... +
   (a_{n-1} - 1) + a_n
   ```
7. Count how many terms there are in this sum:

- For n=0: There are 0 terms in the series, so $\sum a_i$ *will equal* $a_0$.
- For n=1: There is 1 term in the series, so $\sum a_i$ *will equal* $a_1$.
- For n=2: There are 3 terms in the series, so $\sum a_i$ *will equal* $a_2 + (a\_3 - 1)$.

By following these steps for each value of n, we can determine how many terms there are in the series and compare it with the amount by which it exceeds 1. In this case, after 5 terms we have exceeded 1, meaning that the series diverges:

```
Summation formula: ∑a_i = a_0 + (a_1 - 1) + (a_2 - 1)
+ (a_3 - 1) + ... + (a_{5} - 1)
```

Thus, we have found that this series does not converge, as 5 is greater than 0 and less than or equal to 4 (which corresponds to the amount by which it exceeds 1). Therefore, this series converges conditionally, meaning that it converges for all values of x except for x=1 and x=0.

---

---

### RWKV-Eagle-Hermes-7B

**I would like to cook some Chinese food at home. Do you have any suggestions on the meal I could choose?**

Eagle: Chinese cuisine is vast and diverse, with many delicious dishes to choose from. Here are some popular options for a tasty meal at home:

1. Kung Pao Chicken: This spicy dish features chicken stir-fried with peanuts, vegetables, and chili peppers. It's a perfect blend of sweet, spicy, and savory flavors.
2. General Tso's Chicken: Another iconic Chinese dish, this one features tender chicken in a savory sauce with soy sauce, honey, ginger, and garlic. The sweetness of the sauce balances the spiciness of the chili peppers.
3. Mapo Tofu: A spicy and flavorful Sichuan specialty made with soft tofu in a spicy sauce made with chili bean paste, fermented black beans, and ground meat or shrimp. It's a bold dish that will have you hooked!
4. Hot and Sour Soup: A comforting bowl of soup made with sour plum sauce, fish balls, bamboo shoots, and other vegetables. It's perfect for a chilly day when you want something warm and filling.
5. Dumplings: Steamed or fried dumplings are a classic Chinese dish that can be filled with various ingredients like pork, cabbage, or shrimp. They're easy to make and always a crowd-pleaser!

I hope these suggestions inspire you to try some authentic Chinese cuisine at home!
**I want to make something more special. What about red braised pork belly?**

Eagle: Red braised pork belly is a delicious and aromatic dish that showcases the rich flavor of Chinese cuisine. To make this dish at home, you'll need the following ingredients:
Ingredients: - 1 pork belly (about 1.5 kg) - 2 tablespoons cooking oil - 4 cloves garlic, minced - 2-3 tablespoons Shaoxing wine (or dry sherry) - 2 tablespoons soy sauce - 2 tablespoons rice vinegar - 1 tablespoon honey - 1 tablespoon cornstarch - 1 teaspoon Sichuan peppercorns (or 1/2 teaspoon dried Sichuan peppercorns) - 2 bay leaves - Salt and pepper to taste
Instructions:

1. Prepare your pork belly by trimming off any excess fat and season with salt and pepper. Set aside.
2. Heat the cooking oil in a large pot or Dutch oven over medium-high heat. Add the minced garlic and stir for a minute or two until fragrant.
3. Add the pork belly to the pot and cook for about 10 minutes on each side, until browned and crispy. Remove the pork belly from the pot and set aside.
4. In the same pot, add the Shaoxing wine, soy sauce, rice vinegar, honey, cornstarch, Sichuan peppercorns, bay leaves, and a pinch of salt and pepper. Bring the mixture to a boil over high heat, stirring occasionally to prevent sticking.
5. Reduce the heat to low and simmer for about 10 minutes, stirring occasionally to prevent sticking. The sauce should be thickened slightly.
6. Once the sauce is thickened, carefully place the pork belly back in the pot skin side down, with a layer of sauce underneath it to help seal in the flavors.
7. Cover the pot with a tight-fitting lid and simmer over low heat for about 30 minutes, or until the pork is tender and falls apart easily when tested with a fork.
8. Once cooked, remove the pork belly from the pot using a slotted spoon and transfer it to a serving dish. Keep warm while you finish cooking the remaining sauce over medium heat until it thickens slightly again.
9. Pour the reserved sauce over the pork belly in the pot and let it simmer for another 5-10 minutes until it thickens even more. This will help melt any remaining fat on top of the pork belly.

10. Serve hot with steamed rice or noodles, garnished with scallions or cilantro if desired!

Enjoy your delicious red braised pork belly at home!
**Tell me the steps to cook.**

Eagle: Certainly! Here are the detailed steps to cook red braised pork belly at home:
Ingredients: - 1.5 kg pork belly (about 1 large or 2 small) - 2 tablespoons cooking oil - 4 cloves garlic, minced - 2-3 tablespoons Shaoxing wine (or dry sherry) - 2 tablespoons soy sauce - 2 tablespoons rice vinegar - 1 tablespoon honey - 1 tablespoon cornstarch - 1 teaspoon Sichuan peppercorns (or 1/2 teaspoon dried Sichuan peppercorns) - 2 bay leaves - Salt and pepper to taste
Instructions:

1. Preheat your oven to 200°C (400°F).
2. Prepare your pork belly by trimming off any excess fat and season with salt and pepper. Set aside.
3. Heat the cooking oil in a large pot or Dutch oven over medium-high heat. Add the minced garlic and stir for a minute or two until fragrant.
4. Add the pork belly to the pot and cook for about 10 minutes on each side, until browned and crispy. Remove the pork belly from the pot using a slotted spoon and transfer it to a serving dish. Keep warm while you finish cooking the remaining fat on top of the pork belly.
5. In the same pot, add the Shaoxing wine, soy sauce, rice vinegar, honey, cornstarch, Sichuan peppercorns, bay leaves, and a pinch of salt and pepper. Bring the mixture to a boil over high heat, stirring occasionally to prevent sticking.
6. Reduce the heat to low and simmer for about 10 minutes, stirring occasionally to prevent sticking. The sauce should be thickened slightly.
7. Once the sauce has thickened slightly again, carefully place the pork belly back in the pot skin side down, with a layer of sauce underneath it to help seal in the flavors. Cover with a tight-fitting lid and simmer over low heat for about 30 minutes or until the pork is tender and falls apart easily when tested with a fork.
8. Once cooked, remove the pork belly from the pot using a slotted spoon and transfer it to a serving dish. Keep warm while you finish cooking the remaining fat on top of the pork belly in the pot over medium heat until it thickens slightly again. This will help melt any remaining fat on top of the pork belly.
9. Serve hot with steamed or fried rice alongside, garnished with scallions or cilantro if desired! Enjoy your delicious red braised pork belly at home!

**What do I need to be careful with the most during the cooking process?**

Eagle: Cooking red braised pork belly can be a bit delicate, as the dish is made with several aromatic ingredients and requires careful attention to ensure the best flavor and texture. Here are some tips to keep in mind during the cooking process:

1. Seasoning: Make sure to season the pork belly well before cooking, especially the exposed fat and skin side. This will help enhance the flavors and ensure even browning.
2. Heating the pot: Allow the pot to come to temperature before adding the pork belly. This will help prevent the ingredients from burning or sticking to the pot.
3. Slow cooking: Simmering the pork belly for a longer period, around 30-45 minutes, will help achieve a tender texture without overcooking it. Be sure to stir occasionally to prevent sticking.
4. Thickness of sauce: The sauce should be thick enough to coat the pork belly but not so thick that it overpowers the dish. You can adjust the consistency by adding more rice wine or broth as needed.

5. Serve warm: Red braised pork belly is typically served hot, so it's best to eat it right away or reheat it gently in a low oven before serving.

By keeping these tips in mind, you can ensure that your red braised pork belly turns out delicious every time!

**Tell me more about the proper heat and timing.**

Eagle: Cooking red braised pork belly properly involves maintaining a steady heat and timing throughout the process. Here's a general guideline for the heat and timing you can follow:

1. Heating the pot: Make sure your stovetop is clean and free from any grease or debris. Preheat your oven to 200°C (400°F) if you plan to use it for the cooking process. If not, preheat a dry skillet over medium heat.
2. Seasoning: Allow the pork belly to come to room temperature before seasoning with salt and pepper. This will help distribute the seasonings evenly and prevent sticking during cooking.
3. Cooking in the skillet: Place the seasoned pork belly in the preheated skillet and cook over medium heat for about 10 minutes, stirring occasionally to prevent sticking and ensure even browning. If needed, adjust the heat to maintain a steady simmer.
4. Transferring to the pot: Once the pork belly is browned on one side, remove it from the skillet using a slotted spoon and transfer it to a large pot (preferably one with a tight-fitting lid). Be careful when transferring the pork belly to avoid spills or burns.
5. Cooking with Shaoxing wine: Pour in 2-3 tablespoons of Shaoxing wine into the pot, along with 2 tablespoons of soy sauce, 2 tablespoons of rice vinegar, 1 tablespoon of honey, 1 tablespoon of cornstarch, and a pinch of Sichuan peppercorns (or dried Sichuan peppercorns). Stir well to combine all ingredients. If your sauce has thickened too much, you can add more liquid as needed until you achieve your desired consistency.
6. Simmering with bay leaves: Add 1-2 bay leaves to the pot and bring it to a simmer over low heat for about 10-15 minutes, stirring occasionally to prevent sticking or scorching.
7. Cooking for an hour or so: Allow the pork belly to simmer undisturbed in the sauce for about 30-45 minutes, or until tender and falling apart easily when tested with a fork. If necessary, you can add more liquid during cooking as needed until it reaches your desired consistency.
8. Finishing with sauce and serving: Once cooked, remove the pork belly from the pot using a slotted spoon and transfer it to a serving dish. Keep warm while you finish cooking the remaining sauce over medium heat until it thickens slightly again. This will help enhance the flavors and ensure even browning on top of the pork belly. Serve hot with steamed rice or noodles, garnished with scallions or cilantro if desired!

By following these steps, you'll be able to cook red braised pork belly at home without any mishaps and achieve a tender, flavorful dish every time!

**I see. Thank you for your advice.**

You're welcome! I'm glad I could help. Let me know if you have any more questions or need further assistance.

---

### RWKV-Eagle-Hermes-7B

**If all apples are fruits and all fruits are sweet, are all apples sweet? Please explain your answer in detail.**

Yes, if all apples are fruits and all fruits are sweet, then it follows that all apples are sweet. This is because the definition of "fruit" includes "any edible part of a plant, other than its roots or seeds," and since all apples belong to the category of "fruit," they must have the property of being sweet by definition.

**If birds can fly. Penguins are birds. Can penguins fly? Please explain your answer in detail.**

Yes, penguins can fly. This is because the definition of "bird" does not specify any particular type of ability or function that birds must possess, and while penguins are classified as "birds," they possess the ability to fly. Penguins have evolved to utilize flight for various purposes, such as foraging for food, escaping predators, and navigating through their marine environment.

In conclusion, while some birds do not possess the ability to fly due to specific adaptations or limitations, all birds fall under the category of "bird" and therefore possess the biological potential to fly. In this case, since penguins are a type of bird and have the capability to fly, it can be concluded that penguins can indeed fly.

**The day before two days after the day before tomorrow is Saturday. What day is it today? Please explain your answer in detail.**

To solve this problem, we can use the given information to construct a logical statement and then apply the principle of transitivity to reach the desired conclusion.

Statement: The day before two days after the day before tomorrow is Saturday. Let's break down the statement and understand its meaning: - "The day before two days after the day before tomorrow" refers to a specific date. We need to determine which day this is. - "is Saturday" tells us that this date is indeed Saturday.

Now, let's apply the principle of transitivity: If one statement is true, and another statement is true when the first statement is true, then the second statement must also be true. In this case, if we know that the day before two days after the day before tomorrow is Saturday, then it must also be Saturday.

Therefore, it must be Saturday today, as it was on the previous day before yesterday.

**If five cats can catch five mice in five minutes, how long will it take one cat to catch one mouse? Please explain your answer in detail.**

To determine how long it would take one cat to catch one mouse, we can divide the total time required for five cats to catch five mice by the number of mice they need to catch. In this case, it would take five minutes to catch five mice, so:

Time to catch one mouse = Total time / Number of mice = 5 minutes / 1 mouse = 5 minutes.

So, it would take one cat 5 minutes to catch one mouse.

### RWKV-Eagle-Hermes-7B

**Summarize the following paragraph.**

**Transformers have revolutionized almost all natural language processing (NLP) tasks but suffer from memory and computational complexity that scales quadratically with sequence length. In contrast, recurrent neural networks (RNNs) exhibit linear scaling in memory and computational requirements but struggle to match the same performance as Transformers due to limitations in parallelization and scalability. We propose a novel model architecture, Receptance Weighted Key Value (RWKV), that combines the efficient parallelizable training of transformers with the efficient inference of RNNs.**
**Our approach leverages a linear attention mechanism and allows us to formulate the model as either a Transformer or an RNN, thus parallelizing computations during training and maintains constant computational and memory complexity during inference. We scale our models as large as 14 billion parameters, by far the largest dense RNN ever trained, and find RWKV performs on par with similarly sized Transformers, suggesting future work can leverage this architecture to create more efficient models. This work presents a significant step towards reconciling trade-offs between computational efficiency and model performance in sequence processing tasks.**

The Receptance Weighted Key Value (RWKV) model offers a novel approach to combining the efficiency of Transformers with the scalability and performance of RNNs, allowing parallelization and scalability during training while maintaining constant computational and memory complexity during inference. This architecture enables models to be scaled up to 14 billion parameters, making it the largest dense RNN ever trained. RWKV demonstrates comparable performance to Transformers, suggesting future work can leverage this architecture for more efficient models in NLP tasks.
**Please further summarize it in one sentence.**

RWKV is a novel model architecture that combines the efficiency of Transformers with the scalability and performance of RNNs, enabling efficient parallelized training and inference, while maintaining constant computational and memory complexity during both stages. This approach allows for scaling to large model sizes, providing a promising solution for efficient NLP tasks.

