# OpenReview forum: "Eagle and Finch: RWKV with Matrix-Valued States and Dynamic Recurrence"
_colmweb.org/COLM/2024/Conference — COLM_

### Official Review · Reviewer_xmJS · 2024-05-01

**Rating:** 8
**Confidence:** 3
**Ethics Flag:** 1

**Summary:**

This paper provides extensions to RWKV-4 architecture, which is an linear attention based architecture for faster training and inference with limited memory, overcoming the limitations of RNN-like architectures. The contributions of the paper are as follows:

(1) Eagle architecture - improves the earlier architecture with multi-headed valued states, and additional gating mechanism.

(2) Finch architecture - which extends eagle architecture with Lora layers to allow for trainable weights to effectively learn data decay vectors.

(3) Release of multilingual datasets and pre-trained models can benefit the community.

**Questions To Authors:**

1. How well does the proposed changes work for needle in the haystack like long context tasks?

2. What are the reasons for poor performance on MTEB like information aggregation task?

3. What is the performance on code benchmarks? What are the reasons for shortcomings?

**Reasons To Accept:**

1. Paper clearly highlights the proposed extension and they improve upon the original RWKV architecture.

2. Extensive empirical results on various benchmarks (english and multilingual) show the efficacy of the proposed changes.

3. Multimodal experiments in speech and vision showcase the utility of such architectures in modelling long sequences.

**Reasons To Reject:**

1. While there are computational advantage in using extend RWKV architecture, their performance on long context tasks still need more investigations.

2. Exploration of RWKV architectures with instruction tuning needs to be explored.

3. Performance on coding benchmarks leaves room for improvement.

---

> ### Author Rebuttal · Authors · 2024-05-31
>
> > ... instruction tuning needs to be explored.
> Unfortunately at the time of submission we were unable to provide instruction tuned versions. We do have instruction tuned versions available now.
> For example, https://huggingface.co/spaces/devingulliver/subquadratic-llm-leaderboard
> Here you can see RWKV/rwkv-5-world-7b vs EleutherAI/Hermes-RWKV-v5-7B (instruction tuned on Hermes for 4 epochs) where the Hermes instruction-tuned version outperforms the base model across ARC, HellaSwag, MMLU, TruthfulQA, Winogrande, and GSM8K at an average accuracy of 46.42% versus 45.15% for the base model.
>
> > What is the performance on code benchmarks? ...
>
> We have a new result available that shows the improved compression rate of the RWKV World tokenizer versus the Llama2 tokenizer, which includes an experiment on code from the starcoder dataset. All datasets are language subsets from https://huggingface.co/datasets/uonlp/CulturaX with the exception of Code, which uses the dataset https://huggingface.co/datasets/codecomplete/starcoderdata_0.003
>
> | tokenizer | lang | tokens        | codepts       | codepts per token | vocab size | codepts per bit |
> |-----------|------|---------------|---------------|-------------------|------------|-----------------|
> | RWKV      | EN   | 878,861,532   | 3,918,475,074 | 4.459             | 65,536     | 0.2787          |
> | Llama2    | EN   | 1,016,595,271 | 3,918,475,074 | 3.855             | 32,000     | 0.2576          |
> | RWKV      | ZH   | 997,736,792   | 1,056,687,183 | 1.059             | 65,536     | 0.0662          |
> | Llama2    | ZH   | 1,524,486,994 | 1,056,687,183 | 0.693             | 32,000     | 0.0463          |
> | RWKV      | AR   | 1,133,572,680 | 1,765,106,557 | 1.557             | 65,536     | 0.0973          |
> | Llama2    | AR   | 1,569,786,022 | 1,765,106,557 | 1.124             | 32,000     | 0.0751          |
> | RWKV      | HI   | 1,501,237,423 | 1,837,327,906 | 1.224             | 65,536     | 0.0765          |
> | Llama2    | HI   | 1,883,783,695 | 1,837,327,906 | 0.975             | 32,000     | 0.0652          |
> | RWKV      | ES   | 867,595,572   | 3,047,372,943 | 3.512             | 65,536     | 0.2195          |
> | Llama2    | ES   | 938,883,427   | 3,047,372,943 | 3.246             | 32,000     | 0.2169          |
> | RWKV      | Code | 305,141,416   | 1,046,274,579 | 3.429             | 65,536     | 0.2143          |
> | Llama2    | Code | 369,239,882   | 1,046,274,579 | 2.834             | 32,000     | 0.1893          |

---

> > ### Comment · Reviewer_xmJS · 2024-06-05
> >
> > Thanks for the rebuttal. I have read all the reviews and their rebuttals. I would like to keep my score.

---

### Official Review · Reviewer_RBCT · 2024-05-14

**Rating:** 7
**Confidence:** 4
**Ethics Flag:** 1

**Summary:**

This paper introduces Eagle (RWKV-5) and Finch (RWKV-6), novel RNN architectures building upon the RWKV framework.  These architectures utilize multi-headed matrix-valued states and dynamic recurrence mechanisms for improved expressivity while maintaining RNN-like efficiency.  The authors also present a new multilingual dataset, RWKV World v2, and a fast tokenizer to enhance performance on multilingual and code data.  Their experiments demonstrate that Eagle and Finch achieve competitive performance on a wide variety of benchmarks, including language modeling, associative recall, and even multimodal tasks.

**Questions To Authors:**

In the results tables, were the comparisons apples-to-apples for any of the comparisons? It is hard to tell what has the same training data, number of steps, etc. to make a sound judgment on improvements over RWKV 4 and other baselines.

**Reasons To Accept:**

This paper presents an advancement in RNN-based language models with the introduction of Eagle and Finch (RWKV-5 and RWKV-6). The authors demonstrate that these architectures, incorporating multiheaded matrix-valued states and dynamic data-driven recurrence, achieve strong performance on a diverse set of benchmarks, including MQAR and various linguistic tasks. Notably, their performance rivals, and in some cases surpasses, traditional transformer models of similar size, highlighting the potential for creating more efficient models without sacrificing accuracy.

The authors also introduce a new multilingual corpus, RWKV World v2, and a fast tokenizer based on greedy matching, helping to address the need for diverse language representation in language models.  They also publicly release six pre-trained models under open source, trained on this expansive 1.12 trillion token dataset.

**Reasons To Reject:**

The primary concern lies in the lack of detailed analysis and justification for the architectural choices made. The paper states that Eagle and Finch improve upon RWKV-4, introducing features like matrix-valued attention states and data-dependent token shifting. However, the specific benefits of these design decisions are not thoroughly explored. The authors do not provide sufficient evidence or ablation studies to support these claims. Additionally, the paper lacks a comparative analysis of Eagle and Finch against other recently proposed efficient transformer variants and state-space models in an apples-to-apples comparison as far as I can tell.

Complicating comparisons, the paper introduces a new dataset and tokenizer specifically designed for the proposed models, which makes a direct comparison with other models more hazy.

---

> ### Author Rebuttal · Authors · 2024-05-31
>
> > The primary concern lies in the lack of detailed analysis ...
>
> Thank you for the important point about further analysis being desirable. Although data-dependent token shifting is not explicitly ablated in the paper, we do [implicitly] ablate the difference between Eagle (RWKV-5) and Finch (RWKV-6). Data dependent decay is the main difference between the two architectures. We do apply DDLerp to the receptance, key, and value matrices. We found improvements by doing so, which was worth the cost of a small number of parameters.
> We’ve now run an ablation on DDLerp to demonstrate its benefit to the each component. We trained a small 6 layer, d_model=768 Finch model on the 1.6B token minipile dataset at context length 512 and obtained the following final validation loss results:
>
> Finch: 2.91
>
> Finch with DDLerp only on decay: 2.923
>
> Finch with no DDLerp at all: 2.926
>
> Where indicated, we removed only the data-dependent term from the token-shift LERP in these experiments.
> While we do not have specific ablation studies on the other factors like matrix valued states, as other papers have been released at this later date we can now show some references that do. For example, HGRN2 (https://arxiv.org/abs/2404.07904v1) does an ablation study on state expansion via head width in their Section 3.2. xLSTM (https://arxiv.org/abs/2405.04517v1) provides an ablation study on adding matrix memory in their Section 4.2. Both papers show large improvements from adding matrix valued states, with HGRN2 showing improvement up to head size 128.
>
> > Additionally, the paper lacks a comparative analysis ...
>
> Given limited compute resources at the time, the most apples-to-apples comparisons we were able to make prior to submission are shown in Appendix F.1 figures 4 and 5, where we show benchmarks vs total FLOPS used to train each model. We now have apples-to-apples results for models re-trained on Pile and the GPT-NeoX-tokenizer that we will add to camera-ready.
>
> We ran an ablation where we train a 170M RWKV-6 model from scratch on the Pile dataset using GPT-NeoX-20B tokenizer, and compare with Mamba, RWKV-4, and Pythia models at ~160M parameters, all trained on the same dataset and tokenizer. We don't have space to list all new results in this rebuttal, but the average scores for these ablated models on lambada, sciq, record, winogrande, hellaswag, etc. are given below (will add to camera-ready):
>
> RWKV-6 (173M): 52.0%
>
> mamba (168M): 51.3%
>
> pythia (160M): 49.6%
>
> RWKV-4 (169M): 49.7%

---

> > ### Comment · Reviewer_RBCT · 2024-06-07
> > **Reply to Rebuttal**
> >
> > Thank you for answering my questions and all of the additional experiments. I am bumping my score up to 7. I hope these (and the results mentioned in the other rebuttals) are added to the next version of the paper as I think they really do strengthen it.

---

### Official Review · Reviewer_ncRg · 2024-05-16

**Rating:** 6
**Confidence:** 4
**Ethics Flag:** 1

**Summary:**

The paper presents RWKV-5 and RWKV-6, RNN sequence models improved upon the RWKV-4 model of Peng et. al., 2023. Along with the paper, the code for training and inference, as well as a 1.12 trillion-token data set with an emphasis on multilinguality is released. The results include multilingual benchmark results, English-focused benchmark results, using the LM evaluation harness framework of Gao et. al., 2023.  RWKV-5 and RWKV-6 appear to beat Llama-2-7b and a list of other competitive models in the 1B-7B range in the multilingual benchmarks. Both models are worse than Mistral-7B-v0.1 in the English-focused benchmarks. The paper also include additional multimodal experimental results in the Appendix.

**Questions To Authors:**

Can you include Llama-3-7b in the results? Please put the main results of Section 9 in the main body of the paper. Additional results can stay in the Appendix.

**Reasons To Accept:**

The paper represents a substantial amount of work in the space of RNN-based large language models. The results demonstrate RNN-based models can be as competitive as Transformer-based ones. More is to be seen in even larger scales. But the existing experimental results  and insights are already valuable to the research community. The idea of opening sourcing everything should be encouraged.

**Reasons To Reject:**

Some of the claims about the supremacy of the presented models seem to be hyperbolic. For example, the two models are clearly worse than Mistral-7B-v0.1 in Table 4. The claim in the second paragraph of Section 8 is not valid. Since RWKV-5 and RWKV-6 are trained on the RWKV World v2 dataset, it is not clear how much of the improvement comes from the model and how much of it comes from the data. There are not results supporting the tokenizer presented in Section 5.

---

> ### Author Rebuttal · Authors · 2024-05-31
>
> > Some of the claims ... seem to be hyperbolic.
>
> In table 4, we stated that “Eagle and Finch demonstrate exceptionally high capabilities on multi-lingual benchmarks...”. This is a qualified statement about multi-lingual results, which are in Table 3.
> The models in this paper were trained on 1.12T tokens. We have continued training the Eagle 7B model to 2.25T and Finch 7B to 2.5T tokens total with additional data since submission and even on English benchmarks they have now approached Mistral on English with these additional tokens. Specifically, the EagleX 2.25T model surpasses Mistral v0.1 on glue, anli, mnli, mnli_mismatch, and swag.
> The average accuracy across 21 common English-language benchmarks is shown below, showing how Eagle continues to improve towards Mistral as it is trained further:
>
> Eagle 7B World v2 (1.12T tokens): Eng 48.22%
>
> EagleX 7B v1 (1.75T tokens): Eng 53.91%
>
> EagleX 7B v2 (2.25T tokens): Eng 54.95%
>
> Mistral 7B v0.1 (unknown tokens): Eng 56.76%
>
> > The claim in the second paragraph of Section 8 is not valid ... it is not clear how much of the improvement comes from the model and how much of it comes from the data.
>
> We ran an ablation experiment where we train a 170M RWKV-6 model from scratch on the Pile dataset using GPT-NeoX-20B tokenizer, and compare with existing Mamba, RWKV-4, and Pythia models at ~160M parameters, which are all trained on the same dataset and tokenizer. Results show that RWKV-6 still outperforms Mamba by a small margin, and RWKV-4 by a larger gap, indicating that architectural advances indeed contribute to the overall improvement.
>
> We don't have space to list all of these new results in this rebuttal, but the average scores for these ablated models on LAMBADA, SciQ, ReCoRD, WinoGrande, HellaSwag, etc. are given below (will add to camera-ready):
>
> RWKV-6 (173M): 52.0%
>
> mamba (168M): 51.3%
>
> pythia (160M): 49.6%
>
> RWKV-4 (169M): 49.7%
>
> > There are not results supporting the tokenizer presented in Section 5.
>
> We now have a comparison of token count for the dataset https://huggingface.co/datasets/uonlp/CulturaX which we will add to the camera-ready version of the paper and compare across both tokenizers and languages.
>
> > Please put the main results of Section 9 in the main body of the paper. Additional results can stay in the Appendix.
>
> We will add these results into the main paper using the extra camera-ready page.
>
> > Can you include Llama-3-7b ...
>
> Please note that Llama-3-8b was released after our submission.

---

### Decision · Program_Chairs · 2024-07-10

**Decision:**

Accept

**Comment:**

The submission presents two improved RWKV variants for recurrent LLMs, a new tokenizer, and a new training corpus. Reviewers thought that extensive empirical results showed strong performance. However, the paper is weakened by a lack of apples-to-apples comparisons with existing work - e.g. using a new tokenizer and training corpus makes it harder to compare the architecture with previous work. Overall, the paper provides new ideas and experiments that will be useful to the community working on this direction, and I recommend acceptance.